# Satellite Altimetry-based Extension of global-scale in situ river discharge Measurements (SAEM)

Peyman Saemian[1,*], Omid Elmi[1], Molly Stroud[2], Ryan Riggs[3], Benjamin M. Kitambo[4], Fabrice Papa[4], George H. Allen[2], and Mohammad J. Tourian[1]

[1]Institute of Geodesy, University of Stuttgart, Stuttgart, Germany
[2]Department of Geosciences, Virginia Polytechnic Institute and State University, Virginia, USA
[3]Department of Geography, Texas A&M University, College Station, TX, USA
[4]Laboratoire d'Etudes en Géophysique et Océanographie Spatiales (LEGOS), Université de Toulouse, CNES/CNRS/IRD/UT3, Toulouse, France
[*]**Correspondence:** Peyman Saemian (saemian@gis.uni-stuttgart.de)

**Abstract.** River discharge is a crucial measurement, indicating the volume of water flowing through a river cross-section at any given time. However, the existing network of river discharge gauges faces significant issues, largely due to the declining number of active gauges and temporal gaps. Remote sensing, especially radar-based techniques, offers an effective means to this issue. This study introduces the Satellite Altimetry-based Extension of the global-scale in situ river discharge Measurements (SAEM) data set, which utilizes multiple satellite altimetry missions and estimates discharge using the existing worldwide networks of national and international gauges. In SAEM, we have explored 47 000 gauges and estimated height-based discharge for 8 730 of them which is approximately three times the number of gauges of the largest existing remote sensing-based data set. These gauges cover approximately 88% of the total gauged discharge volume. The height-based discharge estimates in SAEM demonstrate a median Kling-Gupta Efficiency (KGE) of 0.48, outperforming current global data sets. In addition to the river discharge time series, the SAEM data set comprises three more products, each contributing a unique facet to better usage of our data: (1) A catalog of Virtual Stations (VSs), defined by certain predefined criteria. In addition to each station's coordinates, this catalog provides information on satellite altimetry missions, distance to the discharge gauge, and relevant quality flags. (2) The altimetric water level time series of those VSs are included, for which we ultimately obtained good-quality discharge data. These water level time series are sourced from both existing Level-3 water level time series and newly generated ones within this study. The Level-3 data are gathered from pre-existing data sets, including Hydroweb.Next (former Hydroweb), the Database of Hydrological Time Series of Inland Waters (DAHITI), the Global River Radar Altimeter Time Series (GRRATS), and HydroSat. (3) SAEM's third product is rating curves for the defined VSs, which map water level values into discharge values, derived using a Nonparametric Stochastic Quantile Mapping Function approach. The SAEM data set can be used to improve hydrological models, inform water resource management, and address non-linear water-related challenges under climate change. The SAEM data set is available from (Saemian et al., 2024) https://doi.org/10.18419/darus-4475 during the review process.

# 1  Introduction

Freshwater is essential for sustaining life on Earth, serving as a critical resource for drinking, agriculture, industry, and ecosystems (e.g., Vörösmarty et al., 2005; Dudgeon et al., 2006; Schewe et al., 2014). Accurate accounting of changes in freshwater
availability is vital for informed water resource management, sustainable development, and addressing the challenges posed by climate change (e.g., Bhaduri et al., 2016; Döll et al., 2016; Garrick et al., 2017; Saemian et al., 2022; Behling et al., 2022; Saemian et al., 2020). To understand global freshwater dynamics, knowledge of river discharge— the volumetric flow rate of water passing through a river cross-section per unit of time—plays a major role (Tarpanelli et al., 2023; Saemian, 2024). Monitoring river discharge across various river systems relies on a global network of national and international gauges. However,
the existing network faces several challenges, particularly stemming from the decreasing number of operational gauges. The decline in monitoring capability is especially pronounced in regions crucial for understanding global water dynamics, such as Africa and Asia (Elmi et al., 2024; Do et al., 2018). Moreover, existing data sets often suffer from delays in accessibility, hampering real-time insights into river discharge dynamics (Riggs et al., 2023). These challenges have prompted the development of alternative methods for monitoring discharge on a large scale.

Unlike discharge, various hydrological and hydraulic variables, such as river water level, river width, and river slope can be measured through remote sensing data (e.g., Smith, 1997; Alsdorf et al., 2007; Tang et al., 2009; Birkinshaw et al., 2010, 2014; Birkett, 1998). By utilizing these observable variables, one can estimate discharge beyond the gauge records through the use of rating curves (Leopold and Maddock, 1953), and even in ungauged locations, by applying algebraic flow laws or hydraulic models. Rating curves are developed by correlating in situ discharge measurements with these river parameters; for example,
water levels derived from satellite altimetry observations (elevation-based rating curves) (Kouraev et al., 2004; Tourian et al., 2013, 2017; Papa et al., 2012, 2010a; Frappart et al., 2015) or river width from satellite imagery (width-based rating curves) (Smith, 1997; Pavelsky, 2014; Elmi et al., 2015; Tarpanelli et al., 2018). These rating curves can subsequently be utilized to estimate discharge solely from remote sensing-based observations.

Several studies have used remote sensing techniques to estimate river discharge on a global scale. One such study by Riggs
et al. (2023) employs remote sensing width observations from Landsat and Sentinel-2 satellites to estimate river discharge across a worldwide network of stations. However, this study only generated rating curves for stations with simultaneous width-discharge data available, limiting the data set to only 2168 gauges. Another notable effort is the Remote Sensing Extension for the GRDC (RSEG) data set, which extends river discharge records for the Global Runoff Data Centre (GRDC) stations by incorporating river width estimates from satellite imagery (Elmi and Tourian, 2023) and water level estimates from satellite
altimetry (Elmi et al., 2024). Similarly, Lin et al. (2023) presented a global implementation of the Bayesian AMHG-Manning (BAM) algorithm and its geomorphologically enhanced variant (geoBAM) using multi-temporal Landsat-derived river widths at over 3,000 river reaches. Additionally, Feng et al. (2021) estimated daily streamflow in 486,493 pan-Arctic river reaches by assimilating over 9 million discharge estimates derived from 155,710 satellite images into hydrological model simulations, providing enhanced insights into the hydrology of the Arctic region and its response to climate change. Despite the advantages
of RSEG in providing extensive spatial and temporal coverage, it still faces limitations with satellite imagery, encountering

difficulties in accurately estimating discharge in narrower rivers and regions with limited satellite data availability due to cloud coverage. Additionally, the data set primarily relies on GRDC stations, further restricting its applicability to areas not covered by the GRDC network.

The objective of this study is to expand and improve the global river discharge records by employing satellite altimetry measurements alongside a comprehensive network of national and international river discharge gauges. Our data set, named the Satellite Altimetry-based Extension of global-scale in situ river discharge Measurements (SAEM) includes:

1. Altimetry-based river discharge estimates along with uncertainty and quality metrics.

2. Water level-discharge non-parametric mapping functions for the defined VSs. These curves model the transformation of water level time series into discharge data using a Nonparametric Stochastic Quantile Mapping Function approach.

3. Water level time series for VSs where good-quality discharge estimates were obtained. We provide SAEM WLs alongside the data set. For the water level time series from Level-3 databases, we only include the specific IDs introduced by the database providers.

4. A catalog of VSs for each gauge that forms the foundation of the SAEM data set. These VSs are defined based on specific criteria and provide information on satellite altimetry missions, SWORD reach ID, distance to the discharge gauge, and quality flags.

SAEM extends the temporal records of inactive stations by combining satellite altimetry and river gauge data, enhancing the availability of global river discharge measurements. This data set supports better water resource assessments and informed decision-making in areas such as sustainable development and climate change adaptation.

## 2 Data sources

### 2.1 SWORD

The SWOT a priori river database (SWORD) developed by Altenau et al. (2021) includes reach boundaries, high-resolution river centerlines, and fixed node locations for river networks worldwide. SWORD contains a consistent topological system and includes crucial hydrological variables such as average surface water elevation, river reach width, and slope at mean river flow for rivers wider than 30 meters. In this study, we utilize the high-resolution river centerlines and reach boundaries provided by SWORD v16. Referred to hereafter as *river reaches*, these components serve as the backbone for our investigation to locate the nearest satellite altimetry's VSs.

### 2.2 In-Situ River Discharge

In this study, we use the daily gauge data curated by Riggs et al. (2023), which includes a comprehensive compilation from both international and national organizations (Table 1). The data set and software package (Riggs et al., 2024) form a global

gauge database (Figure 1). In the data set, when multiple gauges are within approximately 100 meters of each other, the gauge with a longer data record is prioritized to avoid redundancy. All gauge databases utilized in our analysis are publicly accessible through the RivRetrieve software package (Riggs et al., 2024, 2023) except the Chinese Hydrology Project gauge data, which constitutes less than 1% of the gauges considered in this study. Figure 1 demonstrates the number of gauges at each basin. The distribution reveals a higher density of gauges in North America and Europe, while regions such as Africa and parts of Asia have significantly fewer gauges. While gauges can provide data at various temporal resolutions, such as daily or monthly, in the SAEM data set, we have exclusively used daily discharge data from the gauges.

**Table 1.** Gauge data sources used in this analysis.

| Reference | N Gauges | Record Start-End | Date of Access | Reference |
|---|---|---|---|---|
| ArcticNET | 116 | 1913–2003 | 05/2021 | www.r-arcticnet.sr.unh.edu |
| Australian Bureau of Meteorology | 4,340 | 1899–2021 | 09/2021 | www.bom.gov.au/waterdata |
| Brazil National Water Agency | 1,342 | 1920–2021 | 09/2021 | www.snirh.gov.br/hidroweb/serieshistoricas |
| Canada National Water Data Archive | 6,066 | 1860–2021 | 10/2021 | www.canada.ca/en/environment-climate-change |
| Chile Center for Climate and Resilience Research | 501 | 1913–2020 | 09/2021 | https://explorador.cr2.cl/ |
| Chinese Hydrology Project | 141 | 1953–1987 | 09/2021 | (Henck et al., 2010; Schmidt et al., 2011) |
| The Global Runoff Data Centre | 6,614 | 1806–2021 | 09/2021 | https://portal.grdc.bafg.de |
| India Water Resources Information System | 549 | 1960–2020 | 06/2021 | https://indiawris.gov.in |
| Japanese Water Information System | 1,023 | 1954–2019 | 09/2021 | www1.river.go.jp |
| Spain Anuario de Aforos | 1,385 | 1912–2018 | 09/2021 | http://datos.gob.es |
| Thailand Royal Irrigation Department | 126 | 1980–1999 | 09/2021 | http://hydro.iis.u-tokyo.acjp |
| U.S. Geological Survey | 23,634 | 1857–2021 | 09/2021 | https://waterdata.usgs.gov |

### 2.3 Satellite altimetry data

We utilize water level time series from two sources: (1) existing databases that provide Level-3 water level time series, and (2) we generate water levels from satellite altimetry measurements for those stations without any time series in level-3 databases, referred to as SAEM WL. Figure 2 illustrates the distribution of VSs from various data providers. To generate SAEM WL, we follow the methodology described in Tourian et al. (2022) (also described in subsection 3.2), and utilize Level-2 altimetry data described in subsubsection 2.3.2. The following subsections describe the Level-2 and Level-3 altimetry data used in the SAEM data set.

#### 2.3.1 Level-3 satellite altimetric water level from existing databases

Level-3 water level time series are gathered from various databases (listed in Table 2):

1. Hydroweb.Next: The Hydroweb.Next (former Hydroweb) database, accessible at https://hydroweb.next.theia-land.fr/, are specified by LEGOS and computed by CLS on behalf of CNES, Theia, and Copernicus Global Land (Da Silva et al.,

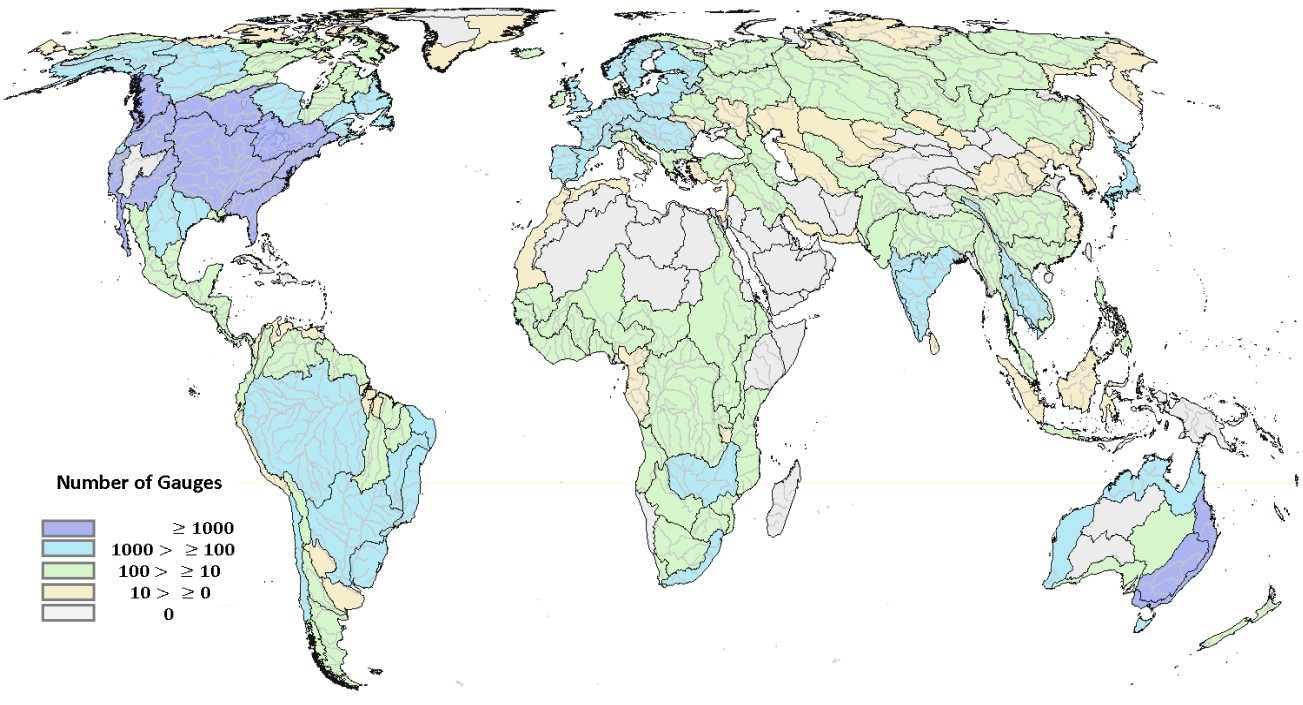

**Figure 1.** Distribution of gauges by basin worldwide. The data set of gauges is sourced from various data centres listed in Table 1.

2010; Normandin et al., 2018). It offers valuable water level time series for more than 24000 VSs globally. The database integrates measurements from satellites including Envisat, Jason-2, Jason-3, Sentinel-3A, Sentinel-3B, and Sentinel-6MF. Hyroweb's processing procedures, as outlined in Da Silva et al. (2010), involve various data sources, including bathymetry maps, Landsat, CBERS-2, SRTM data, and ENVISAT radar images.

2. DAHITI: The Database of Hydrological Time Series of Inland Waters (DAHITI) was developed by the German Geodetic Research Institute at the Technical University of Munich (DGFI-TUM) (Schwatke et al., 2015). Employing an extended outlier rejection and Kalman filter approach, DAHITI integrates cross-calibrated altimeter data from Envisat, ERS-2, Jason-1, Jason-2, TOPEX/Poseidon, SARAL/AltiKa, Sentinel-3, and Sentinel-6MF., yielding comprehensive time series for rivers and lakes globally. DAHITI, as a global database, currently provides 10,758 water level time series distributed across all continents except Antarctica.

3. GRRATS: The Global River Radar Altimeter Time Series (GRRATS) database, spanning from 2002 to 2016, is a globally distributed collection of radar altimeter data from Envisat and Jason-2. This database focuses on ocean-draining rivers with widths exceeding 900 m, encompassing 39 rivers and 1869 VSs. Utilizing an unsupervised method at the virtual station level, GRRATS processed nearly 1.5 million altimeter measurements after quality control. The latest version of GRRATS can be downloaded from https://doi.org/10.5067/PSGRA-SA2V2.

4. HydroSat: This database developed by the Institute of Geodesy, University of Stuttgart, offers geometric quantities of the global water cycle from geodetic satellites. It includes time series and uncertainty estimates for water level from satellite altimetry, surface water extent from satellite imagery, terrestrial water storage anomaly from satellite gravimetry, lake and reservoir water volume anomaly, and river discharge (Tourian et al., 2022). The database is accessible via http://hydrosat.gis.uni-stuttgart.de.

5. Water level over the Congo basin: Kitambo et al. (2022a) utilizes a data set of historical and contemporary river water stages (WSs) and discharge observations, obtained through collaboration with the Congo Basin Water Resources Research Center (CRREBaC). Specifically, the database includes detailed measurements of water levels across the Congo River Basin (CRB). The study has developed a comprehensive water level database, which includes water level records for 1272 VSs. These VSs came from Hydroweb.Next (former Hydroweb) or processed manually using AlTiS (Altimetry Time Series) software.

The primary aim of using these databases is to benefit from the already available Level-3 products and reduce the computational load of processing water level (WL) time series for virtual stations (VSs). In SAEM, we directly utilize the quality-controlled WL time series provided by these databases without any reprocessing or post-processing. Regarding accessibility, all databases are publicly available or upon request. In terms of near-real-time (NRT) data, Hydroweb.Next offers NRT data. When overlaps in VSs were identified between databases, we retained all products in the VS catalog for transparency. The product that passed quality control and achieved better statistical metrics was selected for the final discharge estimates.

**Table 2.** Details of Level-3 water level databases used in SAEM.

| Database | Operated by | # VSs used in this study | Source |
|---|---|---|---|
| Hydroweb.Next | CNES | 24042 | https://hydroweb.next.theia-land.fr/ |
| DAHITI | Deutsches Geodätisches Forschungsinstitut (DGFI) | 9968 | https://dahiti.dgfi.tum.de |
| HydroSat | Insititute of Geodesy, University of Stuttgart | 2036 | http://hydrosat.gis.uni-stuttgart.de |
| GRRATS | Copernicus, European commission ESA, USGS, Amazon Web Services | 1869 | https://blue-dot-observatory.com |
| Congo basin database | (Kitambo et al., 2022b) | 1272 | https://hess.copernicus.org/articles/26/1857/2022/ |

### 2.3.2 Level-2 altimetry Data

For each gauge and any orbit family, if Level-3 water level data is unavailable in the existing databases, we generate water levels using available altimetry missions. To this end, measurements are obtained from a range of satellite altimetry missions, including (1) Envisat, (2) Saral/AltiKa, (3) Jason-1, (4) Jason-2, (5) Jason-3, (6) Sentinel-3A, (7) Sentinel-3B, and (8) Sentinel-6MF. The timeline of the aforementioned satellite altimetry missions is presented in Figure A1.

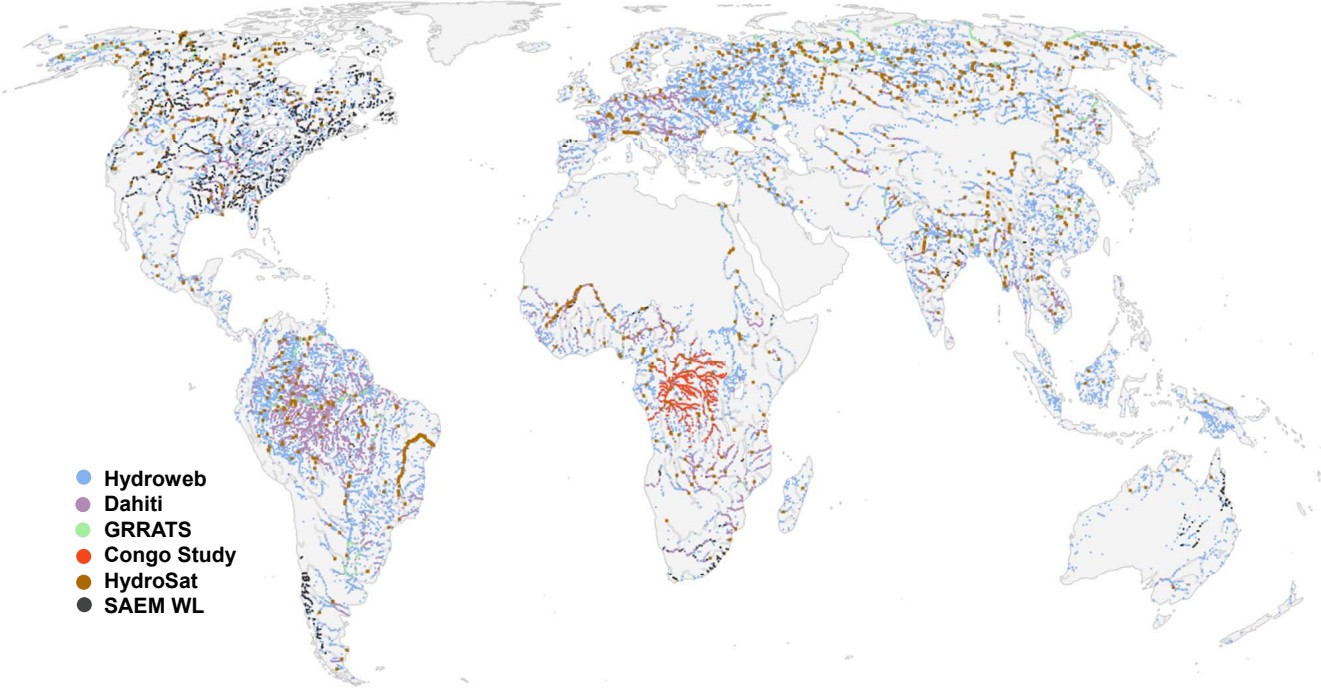

**Figure 2.** Distribution of virtual stations from various data providers used for estimating discharge, including Hydroweb, Dahiti, GRRATS, Congo Study, HydroSat, and SAEM WL.

## 3 Methodology

### 3.1 VS Generation and Selection

A VS in satellite altimetry refers to a specific geographical point where the ground tracks of a satellite altimetry mission intersect with a water body, such as a river (Calmant and Seyler, 2006; Frappart et al., 2006). The process of generating and selecting VSs for the SAEM data set involves integrating multiple sources and several key steps (see Figure 3). Initially, various data sources such as SWORD v16 data, satellite reference orbits, water occurrence from the Global Surface Water (GSW) data set (Pekel et al., 2016), and the geolocation of hydrological gauges are collected. VSs are identified at points where the SWORD river location intersects with the orbital tracks of satellites. After generating all the possible VSs, we need to select the VSs for each gauge located in the vicinity and hydraulically consistent with discharge behaviour. To this end, we have two types of gauges: the ones located on the tributaries of a river system and the ones over the main stem. For gauges located in tributaries, VSs are selected within the same tributary if no dams or reservoirs exist between the gauge and the VS. For gauges on the main river stem, we select those in the mainstream with no intervening dams or reservoirs in between. The presence of dams and reservoirs was determined based on the reach-wise flags provided in the SWORD v16 dataset. This selection ensures that the

VSs accurately represent the hydrological conditions at the gauge locations. The outcome of this procedure is a set of VSs for each gauge, which is one of the SAEM products and is referred to as the *VS catalog*.

To maximize temporal coverage, we include satellite missions from four different orbit families: Envisat series (including ERS1, ERS2, Envisat, Envisat Extended, and Saral/AltiKa), Topex/Jason series (including Topex/Poseidon, Jason 1, Jason 2, Jason 3, and Sentinel 6FM), Sentinel 3A, and Sentinel 3B. It is important to note that after July 2016, SARAL/AltiKa entered a drifting orbit, which provided increased spatial coverage but reduced the temporal consistency at specific locations. For each gauge, at least one available VS is identified from the VS catalog within each orbit category. Next, the availability of water

level time series in Level 3 databases for each VS is checked (see subsubsection 2.3.1 for details about Level 3 databases). If available, the data is collected; if not, water level time series are generated from Level 2 data. The details about the Level-2 data are described in subsubsection 2.3.2, and the methodology to generate the water level is described in subsection 3.2.

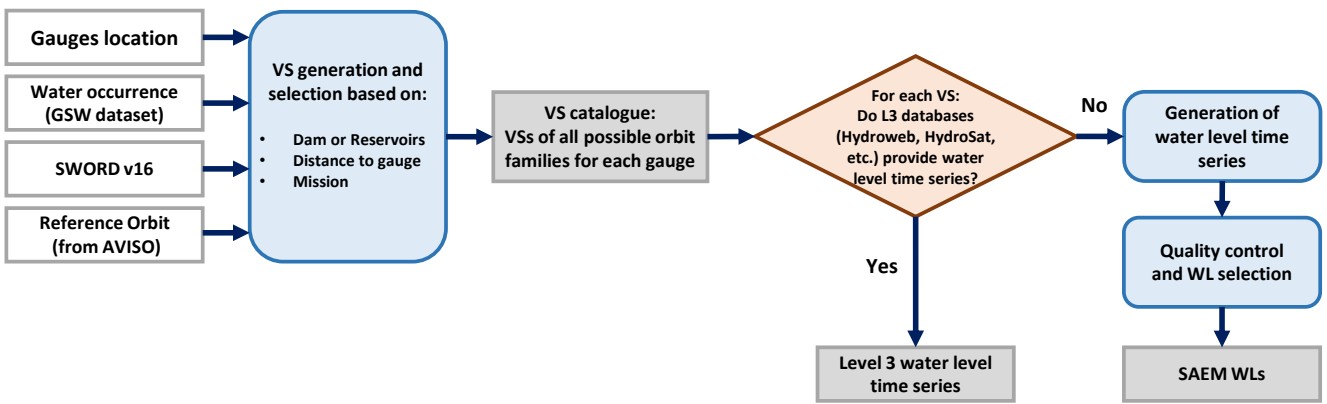

**Figure 3.** Flowchart illustrating the generation of Virtual Stations (VSs), the selection of VSs for each gauge, and the extraction of water level time series using both existing Level-3 databases and newly generated data for this study (SAEM WL).

### 3.2    Generation of water level time series

We attempt to generate water level time series for those VSs that lack in Level 3 databases. For this purpose, for each VS we

first crop all altimetry data with the VS boundaries determined by a static 2 km radius. We then utilize the GSW occurrence map and mask out all data with water occurrence values below 75%. The use of the GSW mask helps maintain the quality of the extracted time series by excluding non-water reflections. Cases with fewer than two valid measurements per epoch are rare (0.6% of all data epochs) and are retained.

The altimetry-derived range measurements $\rho_i$ are obtained from a retracking algorithm. Instead of generating our own

retracking results, we use existing retrackers available in the geophysical data record (GDR). For Envisat and Saral/AltiKa missions, we employ the Ice-1 retracker—a model-free algorithm known for precisely locating the Offset Centre of Gravity of the waveform. In the case of Jason-1 and Jason-3, we utilize the Ice retracker, which employs a similar retracking methodology as Ice-1. For Jason-2, we rely on the results from the PISTACH project and use the Ice-3 retracker. For Sentinel-3A and

Sentinel-3B missions, we utilize the OCOG retracker results applied to the SAR waveforms, while for Sentinel-6MF, we benefit from the MLE retracking algorithm. These selections are based on our experience and analysis of various retracking algorithms across different missions over different inland water bodies.

The range measurements undergo further refinement to account for various geophysical effects. This refinement includes corrections for solid earth tide $\delta\rho_i^{\text{solid}}$, pole tide $\delta\rho_i^{\text{pole}}$, as well as atmospheric path delays such as wet tropospheric $\delta\rho_i^{\text{wet}}$, dry tropospheric $\delta\rho_i^{\text{dry}}$, and ionospheric corrections $\delta\rho_i^{\text{iono}}$. For the wet and dry tropospheric corrections, for all missions, we consistently rely on the models provided by the European Centre for Medium-Range Weather Forecasts (ECMWF), and for the ionospheric correction, we use the results from Global Ionospheric Map (GIM) (Komjathy and Born, 1999). For the pole tide for all missions, the model by Wahr (1985) is used and for the solid Earth tide, the model by Cartwright and Edden (1973).

The corrected range, along with the geoid height $N$, are subtracted from the satellite altitude $H_i$ to derive the orthometric surface water height $h_i$:

$$h_i = H_i - (\rho_i + \delta\rho_i^{\text{dry}} + \delta\rho_i^{\text{wet}} + \delta\rho_i^{\text{iono}} + \delta\rho_i^{\text{solid}} + \delta\rho_i^{\text{pole}}) - N$$

The geoid height $N$ is determined using static gravity field models, specifically referencing XGM2019e (Pail et al., 2018), which has been shown to perform better in regions with limited in-situ gravity data (Zingerle et al., 2020). Achieving uniformity across databases is not feasible, as different Level-3 water level databases, such as Dahiti and Hydroweb.Next, already use different geoid models. However, since all these global models are freely available, users can retrieve geoid heights for their preferred model using lat/lon coordinates and adjust the orthometric height accordingly.

Subsequently, for each time epoch, the representative surface water height $\hat{h}$ within a VS is computed as the median of all $M$ estimated height values $h_i$:

$$\hat{h} = \underset{i \in 1,..M}{\text{median}}(h_i)$$

Accompanying this estimation is the calculation of the standard deviation $\sigma_{\hat{h}}$, which serves as a measure of uncertainty for the estimated water height:

$$\sigma_{\hat{h}} = \sqrt{\frac{1}{M-1} \sum_{i=1}^{M} (h_i - \bar{h})^2}$$

where $\bar{h}$ denotes the mean of water height estimates.

Once the water level time series is generated, we identify and remove outliers through an automated, data-driven outlier identification methodology integrated within an iterative, non-parametric adjustment scheme (Tourian et al., 2022). Finally, quality control is conducted on the generated water level time series to select those with the best quality. This evaluation includes assessing the length of the time series in relation to the theoretical number of observations, which is derived from

the satellite's repeat period and the observation duration. The time series are further checked for statistical characteristics, including the distribution of values, skewness, and variability, to ensure they are representative of water level changes. Outliers are identified and limited to a maximum 10 % to maintain data reliability. Additionally, bias control is implemented by verifying the consistency of mean and median values across segments of the time series. The alignment of the time series with Digital Elevation Model (DEM) information is also assessed to ensure consistency with expected elevations. In total, 3763 WL time series were generated for 1702 gauges across various orbit families, including 1598 from the Envisat orbit family, 990 from the Topex/Jason orbit family, 561 from Sentinel-3A, and 614 from Sentinel-3B. During the quality control process, 632 WL time series were rejected, representing approximately 17% of the total. Rejection rates varied across orbit families, with 494 WL time series rejected from the Envisat orbit family, 34 from Topex/Jason, 64 from Sentinel-3A, and 40 from Sentinel-3B. Those time series passing the quality control, are referred to as *SAEM WL*. The SAEM WL together with water level time series from Level 3 databases, are used as inputs to generate discharge estimates.

### 3.3 Non-parametric Rating Curve Modeling and River Discharge Estimation

Developing an empirical model between the ground- and space-based measurements is the most straightforward approach for extending the discharge record of an inactive gauge station using satellite data. Elmi et al. (2021) developed a nonparametric quantile mapping (NPQM) approach, based on Monte Carlo simulation, for developing a mapping function that transforms remote sensing-based river water level or width time series into discharge estimates. The NPQM method overcomes several limitations of conventional linear regression techniques:

- it does not require simultaneous gauge-based and space-based measurements since the algorithm determines the water height-discharge model by matching the quantile functions,

- it follows a data-driven, nonparametric approach rather than relying on predefined linear or power-law relationships to minimize the possibility of mismodeling

- it provides input-driven discharge uncertainty estimates for each discharge percentile separately rather than relying only on a variance-covariance matrix for model parameters.

The flowchart in Figure 4 describes the procedure of the algorithm.

NPQM performs the following steps to develop the stochastic quantile mapping function:

- generating a stack of river discharge and altimetric water height time series using a Monte Carlo simulation,

- deriving a collection of river water height-discharge mapping functions by matching all possible permutations of the quantile functions of river discharge and height,

- estimating the mean river height-discharge mapping function together with the uncertainty for each percentile,

- evaluating the performance of the derived model by comparing the estimated and measured discharge of the evaluation sample performing a $3\sigma$ test. If available, the evaluation sample consists of simultaneous gauge- and space measurements.

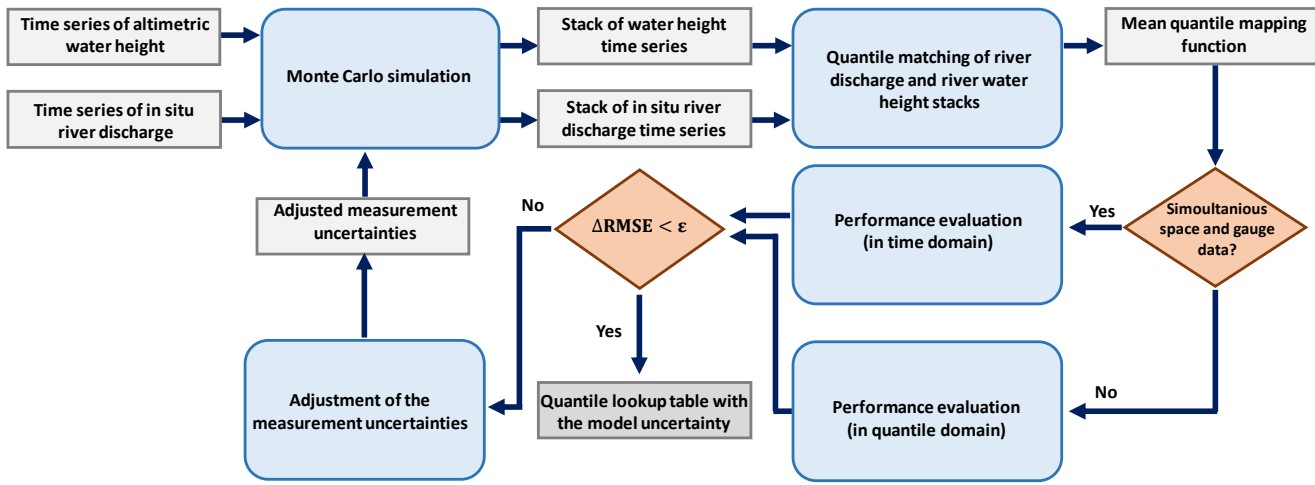

**Figure 4.** Flowchart of the stochastic quantile mapping function algorithm (adapted from Elmi et al. (2021) and Elmi et al. (2024))

Otherwise, measurements from both data sets within the same quantile are included in the evaluation sample. The $3\sigma$ test is a statistical approach that assumes approximately 99.7% of data falls within three standard deviations of the mean in a normal distribution. In this algorithm, the test ensures that the residuals (estimated − measured discharge) remain within this range,

- updating the measurement uncertainties with respect to the result of the $3\sigma$ test, scaling them accordingly to maintain consistency in the iterative process,

- terminating the algorithm if the root mean square error (RMSE) from the previous step does not change significantly, otherwise, the algorithm returns to the first stage.

In the initial iteration, the algorithm considers a multiplicative uncertainty of 10% of the signal for the input time series. This decision is due to the lack of available uncertainty estimates for the gauge discharge dataset and the inconsistent definitions of uncertainties across altimetric water level databases. As the algorithm progresses, it refines its estimates by updating the measurement uncertainties at each iteration. This iterative process continues until the termination condition is met; however, a maximum number of iterations is also set to ensure the algorithm converges within a predefined limit.

Through the procedure, the algorithm generates a stack of quantile mapping functions by propagating the input measurements based on their respective uncertainties. The distortion observed in the collection of mapping functions illustrates the model's accuracy in estimating discharge at various percentiles. The discharge estimation model uncertainties are later exploited to obtain the uncertainty of the RS-based discharge estimates. Once the model is developed, the discharge value, along with its associated uncertainty, can be estimated using solely the predictor variable. A detailed discussion of the uncertainty propagation and the robustness of the NPQM method can be found in Elmi et al. (2021) and Elmi et al. (2024). These stud-

ies explore how uncertainties are handled through the iterative quantile mapping process and their implications for discharge estimates.

## 3.4 Quality Control of the Estimated Discharge

After employing the NPQM to generate discharge estimates, we implement a quality control process to ensure the reliability of these estimates (Figure 5). This process combines statistical tests and visual inspections to identify and exclude low-quality or anomalous data. We have two different cases: (1) where simultaneous time series for discharge and water level are available, and (2) where simultaneous data is not available. For case 1, the initial assessment involves calculating the Kling-Gupta Efficiency (KGE) (Kling et al., 2012) between the in-situ and the estimated height-based discharge; KGE values range from

$-\infty$ to 1, where 1 represents a perfect match between observed and estimated discharge while values below $-0.4$ indicates no skill beyond the mean discharge. A KGE greater than 0 is considered acceptable, as this threshold has been found to produce quality hydrographs based on our experimental analysis. Additionally, the Kolmogorov-Smirnov (KS) (Lopes et al., 2007) test compares the distribution of estimated and measured discharges. The KS test helps determine if there is a significant difference between these distributions. We expect that if the distributions are similar, it indicates that the height-based estimates are

comparable to the gauge measurements. Only those datasets that pass the KS test are retained for further analysis. In cases where simultaneous data are unavailable (Case 2), first, we check the KGE (between the mean monthly discharge of in-situ and estimated data) to be greater than $-0.4$, which indicates an improvement over using the mean observed river discharge. We then assess using the Shapiro-Wilk (SW) test (Shapiro and Wilk, 1965) to check the normality of the difference between the mean monthly discharge of in-situ and estimated, and only those passing the tests are retained. Subsequently, the KS test determines

if the estimated and measured discharge values follow the same distribution. Beyond these statistical tests, a visual inspection is conducted to detect anomalies not captured by quantitative methods. This inspection focuses on identifying unusual long-term patterns, significant variations or anomalies (sudden spikes, drops, or erratic fluctuations), and outliers or extreme values. In such a quality control procedure, about 21860 cases are rejected, ensuring that only reliable and accurate discharge estimates are included in the final data set, thereby enhancing the robustness and credibility of our results.

## 4 Products

The SAEM data set offers a multi-faceted perspective on river system dynamics, combining raw observational data with carefully derived products. This section outlines the key components of the data set, including the Catalog of VSs, Altimetric Water Level Time Series, River height-discharge mapping functions, and Discharge estimates with Uncertainty.

## 4.1 Catalog of Virtual Stations

This catalog comprises VSs selected based on predefined criteria, as detailed in subsection 3.1. Each VS is characterized by coordinate information (latitude and longitude), a unique identifier, and the reach ID from the nearest reach in the SWORD data set. Information related to satellite altimetry, such as the satellite and ground track number, is also included. Two essential

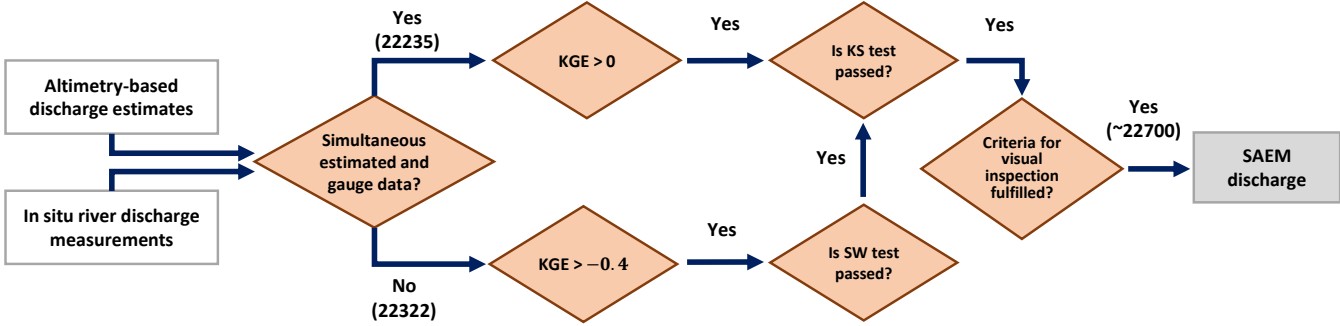

**Figure 5.** Flowchart of the quality assessment procedure for SAEM discharge. We calculate the KGE for non-simultaneous cases between the long-term monthly mean of in-situ data and estimated discharge. The KS test refers to the Kolmogorov-Smirnov test, and the SW test refers to the Shapiro–Wilk test.

flags (flag_wl, and flag_dis) denote the quality of the water level and the feasibility of generating discharge from each VS. This catalog serves as a foundation for subsequent products.

## 4.2 Altimetric Water Level Time Series

For VSs with accepted discharge records, we provide the water level time series generated specifically for this study (SAEM WL). Moreover, for existing Level-3 water level databases, we include the specific IDs introduced by the database providers. The origin of water level data is explicitly indicated in the 'provider' attribute, differentiating between externally sourced Level-3 products and internally generated SAEM data. Although SAEM provides mono-mission discharge estimates using non-parametric rating curves, it is important to acknowledge inter-mission biases that may arise when comparing or combining water level time series (WL TS) from different satellite missions. Such biases, resulting from differences in satellite orbits, calibration, and instrument characteristics, can impact the continuity and consistency of long-term WL TS. Additionally, the use of different retrackers can also introduce biases. In SAEM, the water level time series are specific to the retrackers and processing setups used for each mission. Users who aim to build long-term WL TS by combining Level-3 data from multiple missions should account for these biases to ensure meaningful comparisons.

## 4.3 Height-discharge Mapping Functions

The height-discharge mapping functions represent the intermediary step in transforming water level time series into discharge data. This product includes non-parametric quantile mapping functions specifically for VSs where the final discharge data achieved at least a minimum quality. The inclusion of rating curves allows users to generate discharge using their own water level time series. Additionally, users can estimate uncertainties for their discharge computations using the uncertainty lookup tables provided alongside the rating curves, which are derived from the standard deviation of the Monte Carlo simulations.

## 4.4 Discharge with Uncertainty

The main product in the SAEM data set is the discharge time series together with the uncertainty estimates for gauges that successfully passed the quality control assessments. Each epoch of the time series is associated with a VS identifier stored in a separate vector called VS_id. Additionally, information from the quality control process is embedded in this product, providing insights into the reliability and accuracy of the derived discharge data.

## 5 Results and Validation

In the SAEM data set, we have monitored around 47 000 gauges worldwide in which 8 730 gauges meet the requirements discussed in subsection 3.4 and are included in the SAEM. By implementing our height-based rating curve method across 8 730 gauges, we contribute 1 048 303 day epoch discharge observations to global records. Of these, 614 155 observations extend the gauge records beyond their original periods, while 434 148 observations fill historical gaps in the data. In total, we generated approximately 25,000 orbit-family-based discharge cases, of which about 45% were from cases with simultaneous in-situ discharge and water level data available. Out of the 6,310 gauges not included in SAEM, approximately 22% (1,405 gauges) were within 12 km of a SWORD reach, with the remaining 78% excluded primarily due to greater distances. Of the gauges near SWORD reaches, 48% (671 gauges) had water level (WL) data either from Level-3 databases or SAEM WL. For these gauges, the Non-Parametric Quantile Mapping (NPQM) method failed to converge for any orbit family in 10% of cases (69 gauges), while it successfully converged for at least one orbit family in 90% of cases (602 gauges). Among the 602 cases ultimately rejected, 44% were excluded during visual inspection (263 gauges), and 56% failed statistical thresholds (339 gauges). Figure 6 shows the discharge network across different continents. The gauges are divided into four categories: those for which discharge was successfully estimated and passed visual inspection (SAEM), those for which discharge estimation did not pass our visual inspection or the gauge was too far (>12 km) from the nearest reach of SWORD (Rejected), those without sufficient water level data or no VSs catalog (No WL data), and those with a mean daily discharge below 10 m$^3$/s. In each continent, the share of each group of gauges is shown in rings and their associated color. The outer ring represents the proportion of discharge volume, while the inner ring indicates the proportion in terms of the number of gauges.

The overall analysis shows that despite the selective nature of the gauge inclusion process, which reduced the number of gauges, the most significant gauges in terms of discharge volume are retained. For instance, Africa, Asia, and South America show that the selected gauges account for almost 100 % of the total discharge. Even continents with fewer selected gauges, like Australia and North America, maintain high coverage of total discharge volume (91 % and 98 %, respectively). The SAEM data set, which includes gauges that passed visual inspection, still covers a substantial portion of the total discharge, such as 72 % in Africa, 89 % in Europe, and 85 % in North America. The highest percentage of the total discharge coverage is achieved over South America (92 %) followed by Asia and Europe (each 89 %). The minimum portion of the estimated discharge for the total discharge has happened in Australia (50 %), which is due to the high level of rejection in the process of the SAEM data set. Additionally, the majority of the gauges (83 %) in Australia have a long-term mean monthly discharge below 10 m$^3$/s, meaning they were not selected in the first round of the SAEM data set.

Figure 7 compares the estimated discharge to in-situ discharge measurements across a selection of gauges that represent various continents and data centers. For the gauges with simultaneous data in both SAEM and in-situ records, scatter plots and the corresponding correlation coefficients are provided. In three selected cases where simultaneous data were unavailable, the comparison is presented through monthly mean discharge values. The selected gauges include cases where in-situ data ceased but discharge estimation continued in the SAEM data set (cases 1 to 10). Additionally, there are cases with minimal to no

recent in-situ data, yet the SAEM data set successfully estimates discharge for these periods (as seen in cases 12 to 14). The results show a high correlation between SAEM estimates and in-situ measurements, with correlation coefficients exceeding 0.76, indicating good performance of the SAEM data set. In cases without simultaneous data, the distribution of discharge shown by the gauge is accurately captured and reflected in the estimated discharge. The high levels of correlation demonstrate the reliability of SAEM in estimating discharge accurately, even in the absence of continuous in-situ data.

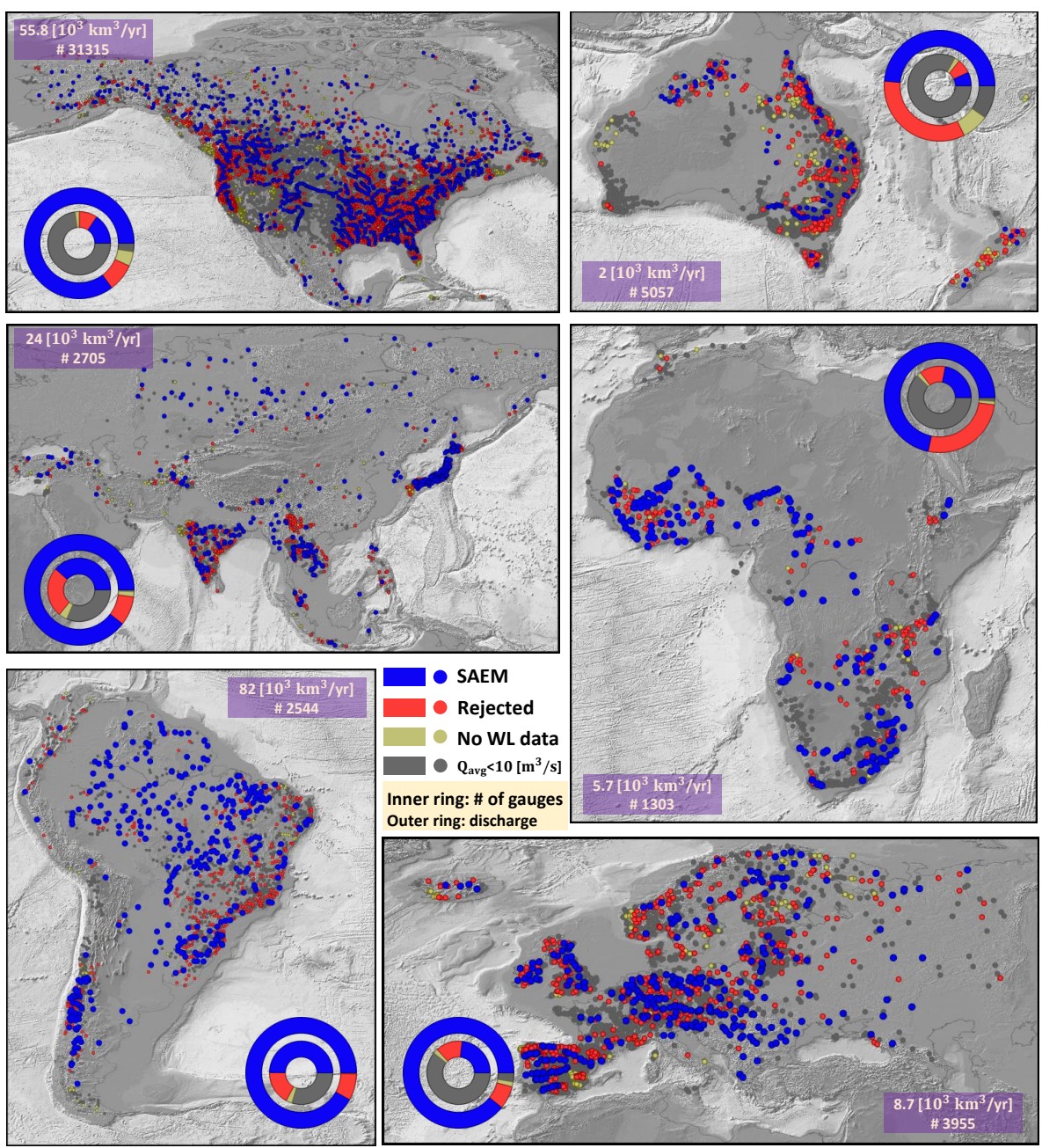

**Figure 6.** Distribution of discharge gauges across different continents categorized based on data availability and discharge estimation status. The outer rings represent the percentage of total discharge volume captured by each category, while the inner rings show the percentage based on the number of gauges. Categories include gauges with successful discharge estimation (SAEM), rejected estimates, insufficient water level data, and gauges with low mean daily discharge or insufficient data duration.

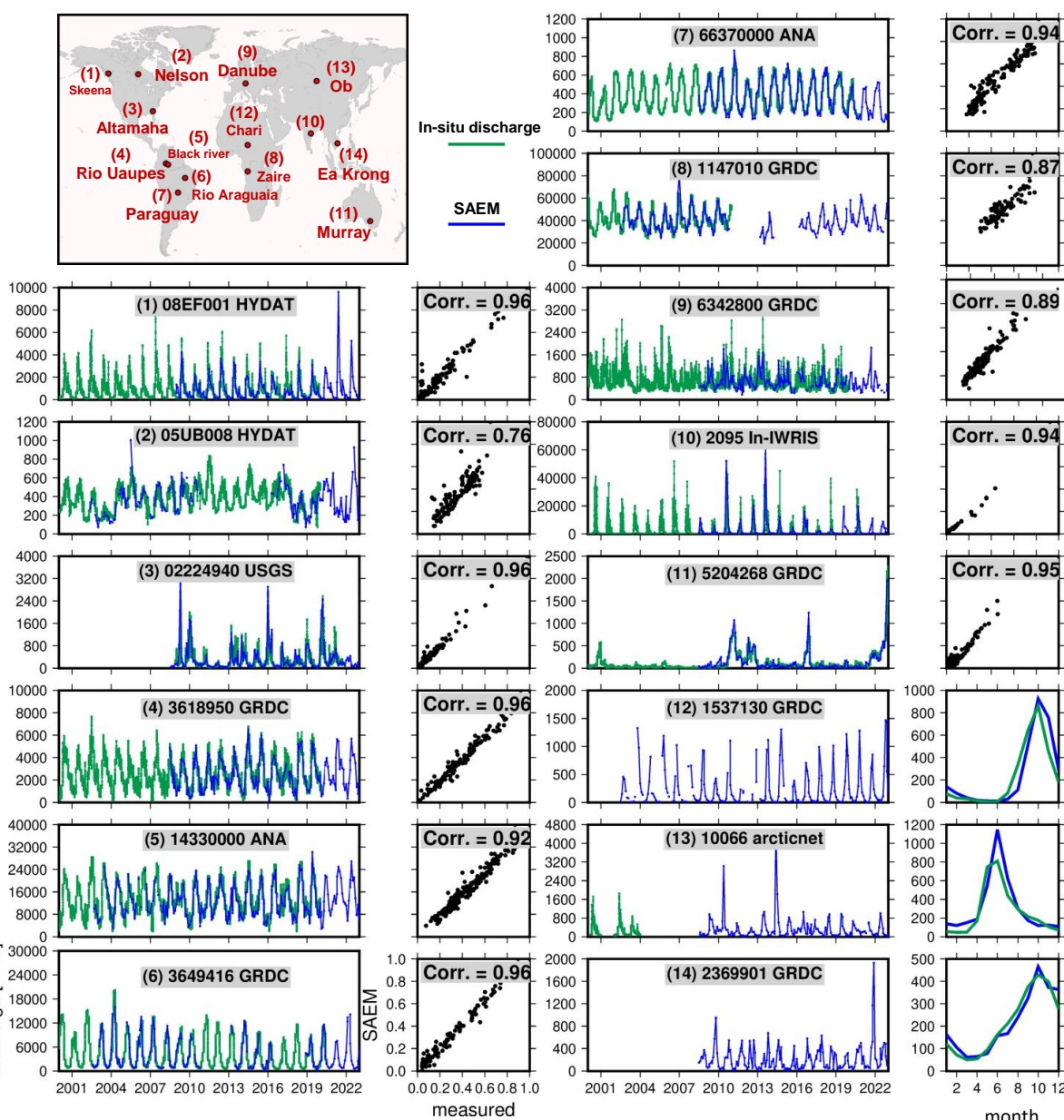

**Figure 7.** Comparison of estimated discharge from the SAEM data set with in-situ discharge measurements across selected gauges from various continents and data centers. For cases 1 to 11, where simultaneous data were available, the time series plots are shown alongside scatter plots and their corresponding correlation coefficients (displayed to the right of each time series plot). For cases without simultaneous data between SAEM and gauged discharge (cases 12, 13, and 14), the monthly means are compared. It should be noted that in the scatter plots, the values are normalized by the maximum discharge value among the simultaneous gauge and SAEM data sets.

We have assessed the performance of the estimated discharge in the SAEM data set against the gauged discharge. Figure 8 presents a detailed comparison of SAEM discharge estimates with in-situ discharge measurements for gauges with simultaneous data, covering approximately 57% of the SAEM data set. The first row showcases the spatial distribution of the Kling-Gupta Efficiency (KGE) and the correlation coefficient (Corr.), with color gradients indicating the performance of the SAEM estimates across various locations. The second row highlights the spatial distribution of the Normalized Root Mean Square Error

(NRMSE) on the left and a scatter plot of the mean daily discharge versus KGE on the right. The NRMSE is normalized using the standard deviation of the observed discharge to ensure comparability across gauges. The bottom row provides cumulative distribution functions (CDFs) for KGE, Corr., and NRMSE metrics.

The spatial plots reveal notable regional patterns in the performance of the SAEM estimates. High KGE and correlation values are predominantly observed in North America, Europe, and parts of South America, indicating that the SAEM model

performs well in these regions. Conversely, regions like parts of Africa and Asia show more variability in performance, with some gauges exhibiting lower KGE and higher NRMSE values. This pattern may be attributed to the density and quality of the in-situ data available in these regions, as well as regional hydrological complexities that may impact the accuracy of the SAEM estimates. The distribution of gauges with simultaneous data is denser in North America, South America, and Europe while being more sparse in regions like Africa and parts of Asia.

The overall good accuracy of the SAEM estimates is evident from the CDF plots. Our findings reveal median values for the KGE, Correlation Coefficient, and the NRMSE of 0.48, 0.64, and 18%, respectively. Moreover, we have evaluated the estimated discharge against in-situ observation for each orbit family (Figure A2). Sentinel 3A stands out with the best overall performance, achieving a median KGE of $0.44$, followed by the Envisat series with a median of $0.35$. The KGE distribution highlights Sentinel 3A's consistent superiority, particularly at higher performance levels. Meanwhile, the correlation analysis

further supports the strength of Sentinel 3A and Envisat, which both show higher correlation coefficients compared to Sentinel 3B (median KGE $= 0.27$) and Topex/Jason (median KGE $= 0.26$). The NRMSE panel reveals that Sentinel 3A and Envisat not only have lower error rates but also maintain tighter distributions, while Topex/Jason exhibits the widest spread in error, indicating less consistency. Several factors influence the difference in performance between Sentinel-3A and Sentinel-3B. Besides the difference in their orbit, Sentinel-3A has been operational since 2016, providing a longer data record compared to

Sentinel-3B, which began in 2018. Additionally, there are significantly more cases with simultaneous water level and discharge data for Sentinel-3A compared to Sentinel-3B (4483 vs 2369). For the Topex/Jason orbit family, the global quality is influenced by the older missions, which may exhibit higher uncertainty due to differing sensor capabilities and retracking algorithms. Based on Figure A2, the Topex/Jason orbit family shows a lower correlation compared to the Envisat orbit family but performs better in terms of NRMSE. This indicates that while the Jason series achieves smaller errors in discharge magnitude, its ability

to capture temporal variability is slightly weaker.

## 5.1   Comparison of discharge time series with existing data sets

We compare the SAEM discharge time series against four existing data sets, namely the Remote Sensing-based Extension for the GRDC (RSEG) (Elmi et al., 2024), the data set developed by Riggs et al. (2023), the global width-based data set

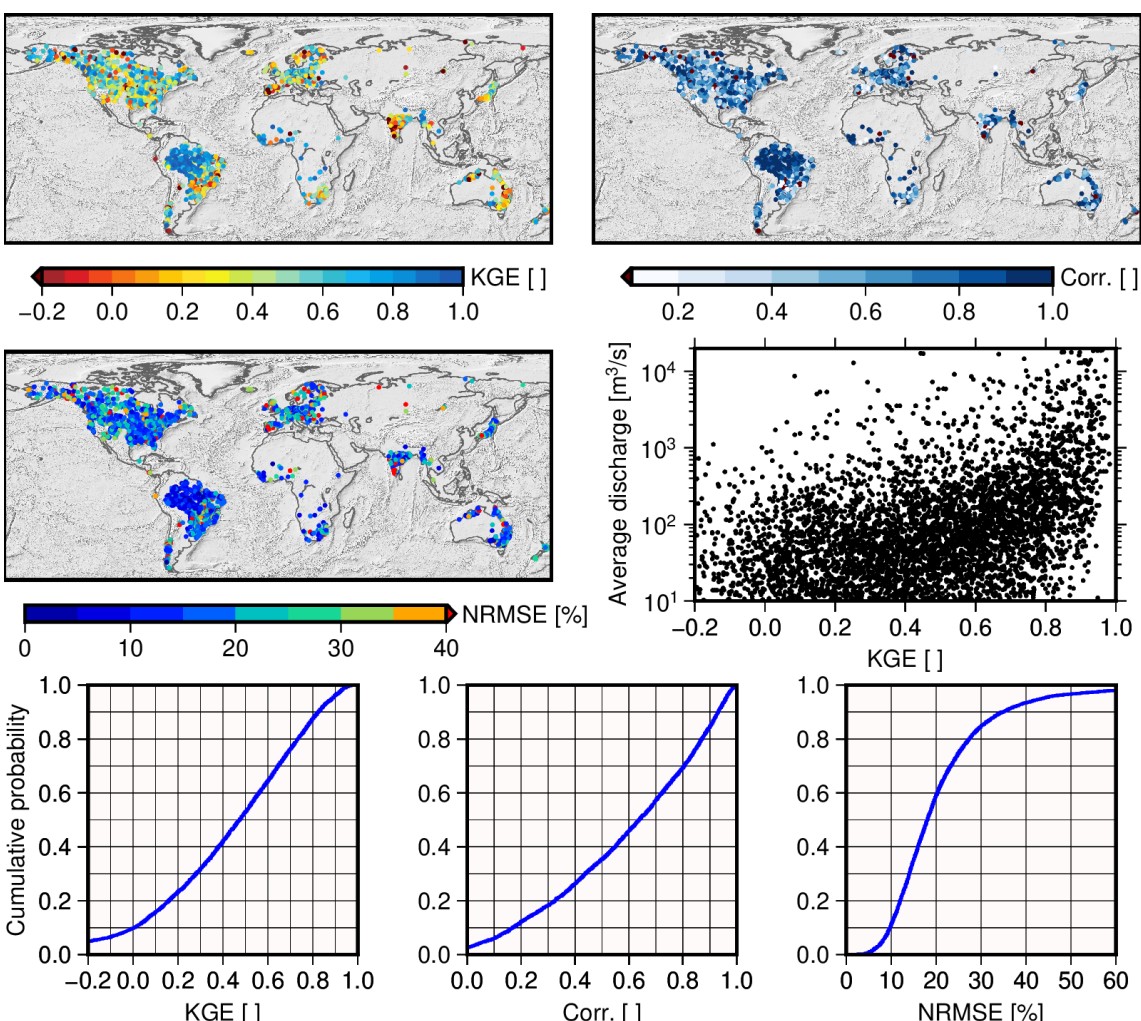

**Figure 8.** Global performance of the SAEM discharge estimates compared to in-situ measurements. The first row displays the spatial distribution of the Kling-Gupta Efficiency (KGE) and correlation coefficients (Corr.) for gauges with simultaneous data. The second row shows the spatial distribution of Normalized Root Mean Square Error (NRMSE) and a scatter plot of mean daily discharge versus KGE. The bottom row presents cumulative distribution functions (CDFs) for KGE, Corr., and NRMSE, illustrating the overall accuracy and reliability of the SAEM data set across different regions.

developed by Lin et al. (2023) (RSQ), and data from ESA Climate Change Initiative (CCI) River Discharge project (Gal et al., 2024) (hereafter simply CCI). The RSEG data set includes monthly discharge time series for 3377 GRDC gauges worldwide, utilizing river water level estimations from satellite altimetry observations, river width estimations from satellite imagery applying the algorithm introduced in (Elmi and Tourian, 2023), or a combination of both in some gauges. Riggs et al. (2023) benefited from river width observations from Landsat and Sentinel-2 satellites and filled the missing records at 2168 gauges worldwide. The RSQ dataset Lin et al. (2023) includes 3078 gauges globally, where river discharge was estimated

using both the Bayesian AMHG-Manning (BAM) algorithm and its geomorphologically-enhanced variant (geoBAM). Finally, CCI has developed parametric height-based rating curves for 46 gauges to extent the river discharge measurements. CCI aims to create over 20-year climate data records of river discharge for selected basins using satellite remote sensing (altimetry and multispectral images) and supporting data. It also serves as a proof-of-concept for a potential River Discharge Essential Climate Variable (ECV) product to meet Global Climate Observing System requirements (for more information visit https://climate.esa.int/en/projects/river-discharge/).

Figure 9 presents the results from the comparison of SAEM with RSEG and Riggs et al. (2023). Between the SAEM and RSEG data sets, there are 2259 gauges in common. We can divide these gauges into three categories based on their input remote sensing data in RSEG: (1) 91 gauges are based solely on height, (2) 1745 gauges are based solely on width, and (3) 423 gauges use a combination of both in the RSEG data set. The distribution of the RSEG (in grey) gauges and the common gauges (in green) is shown in Figure 9 (a). To compare these common gauges with in-situ measurements, we first up-scaled the SAEM time series to a monthly time scale, as the RSEG data set provides only monthly estimations. We then included only the periods where at least 24 values were available for all three data sources: SAEM, RSEG, and in situ. Finally, we ended up with 938 gauges that met this condition. The CDF of the KGE for these gauges from SAEM and RSEG data sets is displayed in Figure 9 (d). The results indicate a slightly better performance of the SAEM data set for the common gauges, with an average improvement of about 0.1 KGE. The slightly better performance of the SAEM data set compared to RSEG may be attributed to the superior sampling frequency of altimetry compared to satellite imagery, particularly in the high latitude regions, where cloud coverage often limits the effectiveness of satellite imagery. Furthermore, as shown in the Figure 9 (a), there are over 1 100 gauges included in the RSEG data set that are not part of SAEM, particularly in Asia and Siberia. This suggests that RSEG and SAEM can complement each other, offering a more comprehensive data set with about 10 000 gauges for monitoring global river discharge.

With the data from Riggs et al. (2023), we have 1972 gauges (out of 2 168) in common. The distribution of these gauges is shown in Figure 9 (b). All the input in Riggs et al. (2023) is based on the width estimation from satellite imagery (Landsat and Sentinel-2). To conduct the comparison, we utilized the width-based discharge time series estimated by Riggs et al. (2023). First, we upscaled both data sets to a monthly scale. We then selected gauges that had at least 24 values across all three data sets: Riggs et al. (2023), SAEM, and in-situ measurements. Three metrics namely Correlation Coefficient, NRMSE, and KGE values were calculated for these gauges, resulting in a final set of 1 362 gauges. The CDF of the KGE for these 1 362 gauges, comparing Riggs et al. (2023) with in-situ data and SAEM with in-situ data, is displayed in Figure 9 (e). Overall, SAEM demonstrates a slightly better performance, with an average improvement of approximately 0.15 KGE ($\sim 0.15$ Corr and $\sim 5\%$ NRMSE). This improvement can be attributed to SAEM's use of altimetry data, which offers more representative measurements with a better sampling compared to satellite imagery, particularly in regions where cloud cover impedes satellite measurements.

Among the 3078 gauges in the RSQ dataset, 1580 are common with the SAEM dataset, as shown in orange in Figure 9 (c). It is important to note that discrepancies in gauge naming systems affected the matching process for approximately 1000 stations, making it difficult to identify common gauges based solely on their coordinates. Therefore, only gauges with matching names

in both RSQ and SAEM datasets were included in the comparison. Both datasets were first upscaled to a monthly scale. For approximately 800 of these common gauges, at least 24 simultaneous data points were available from SAEM, RSQ (BAM and geoBAM), and in-situ observations, enabling a direct comparison of discharge estimates. The CDF of the KGE values for these 800 gauges, comparing RSQ with in-situ data and SAEM with in-situ data, is displayed in Figure 9 (f). Overall, SAEM demonstrates better performance, with an average improvement of approximately 0.15 in KGE compared to RSQ-geoBAM (and 0.35 compared to RSQ-BAM). This improvement is primarily attributed to a reduction in bias, as evidenced by a 20% improvement in NRMSE for RSQ-geoBAM ($\sim 60\%$ for BAM), while the improvement in correlation was negligible. The better performance of SAEM compared to RSQ (both BAM and geoBAM) can similarly be attributed to its use of altimetry data, which provides more representative and reliable measurements with improved sampling, particularly in regions with persistent cloud cover that limits the effectiveness of satellite imagery. Notably, geoBAM outperforms BAM slightly, consistent with findings in Lin et al. (2023).

Due to the thorough care given to obtaining rating curves and river discharge time series within the CCI project (Gal et al., 2024), we use its results as a benchmark for the quality assurance of our product. Out of 46 gauges in CCI, 36 gauges are in common with SAEM. Overall our results agree well with those from CCI. Figure 10 illustrates the comparison between the SAEM data set and the CCI project for the gauge 1134900 of GRDC over the Niger River. The bottom-left plot compares the water level time series from the four mission series in SAEM with those from the CCI River Discharge project. The scatter plot in the bottom right shows a strong correlation between the water levels in SAEM and CCI. This high correlation is because, for most of the time series, both data sets have used similar water level measurements to generate their respective discharge estimates.

The top-left plot compares discharge time series from four mission series in SAEM with those obtained from the CCI project and in-situ measurements. The SAEM data includes contributions of water level from Sentinel 3A (Dahiti), Sentinel 3B (Hydroweb), Envisat (Dahiti), and TOPEX/Jasons (Hydroweb). The CCI, on the other hand, relies primarily on hand-processed time series using the Altis software, particularly for Niger time series. The gauged discharge is also shown in green but only covers till 2001. The discharge time series from both SAEM and CCI exhibit similar patterns. As shown in the top-left plot with bar plots, the absolute difference between CCI and SAEM is, on average, below 80 m³/s ($\sim 3\%$ of the range of discharge), with the maximum difference observed in the Sentinel-3A and 3B series and the minimum difference in the TOPEX/Jasons series. In the top-right plot, the rating curves derived from each of these time series are compared. SAEM uses a non-parametric approach proposed by Elmi et al. (2021), whereas the CCI employs a parametric approach and uses a global optimization algorithm based on the Monte Carlo Markov Chain and Bayesian framework proposed by Paris et al. (2016) to develop the rating curves. The nonparametric quantile mapping functions from SAEM align well with the rating curves from CCI.

## 5.2 Applications

The products in the SAEM data set can be used in various applications in water resource management, monitoring, and climate change studies. For example, the rating curves produced by this study can be used together with operational satellite altimetry missions (Sentinel-3 and Sentinel-6MF) to improve access to Near Real-Time (NRT) discharge estimates. Moreover, SAEM

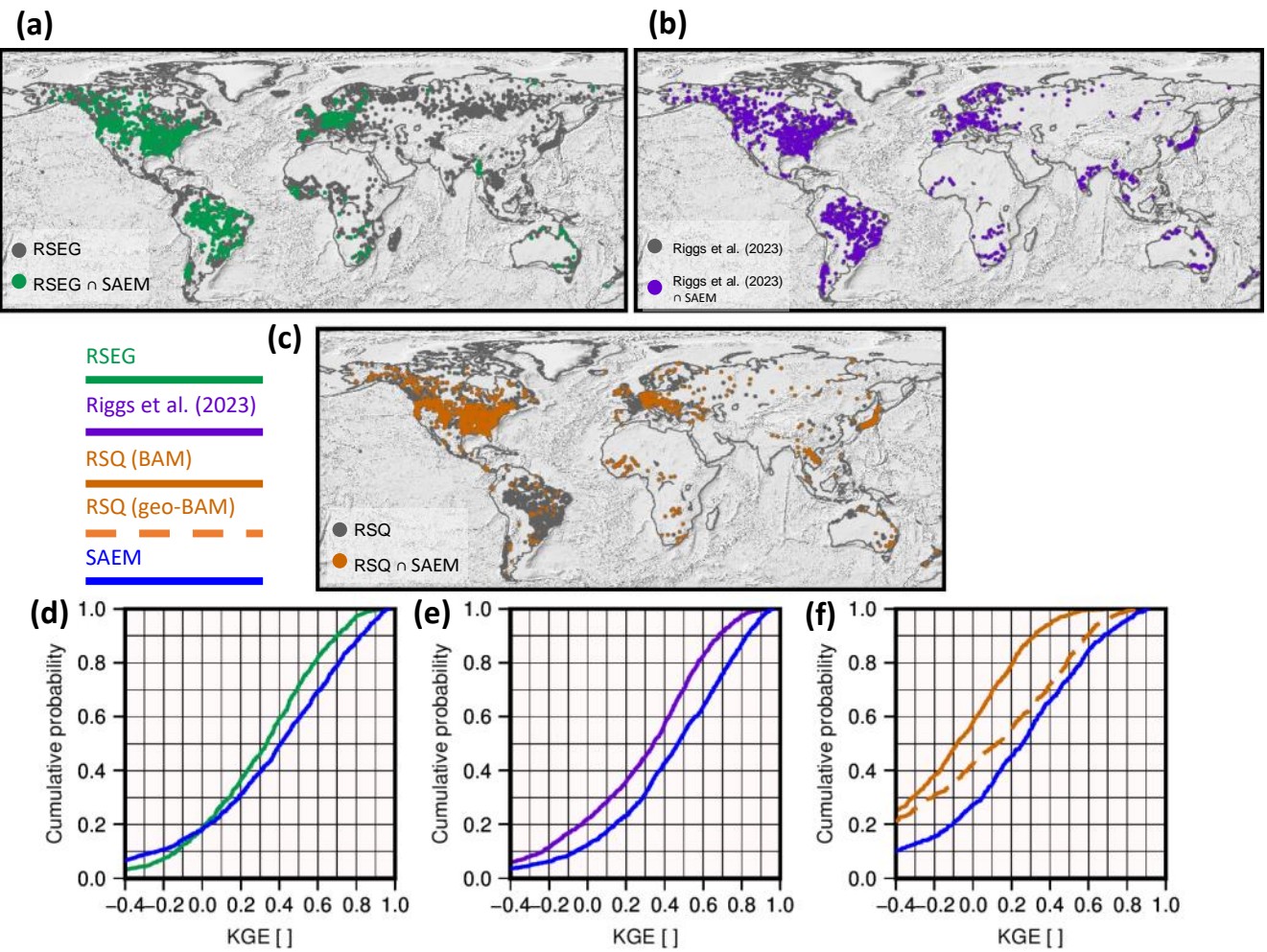

**Figure 9.** Comparison of the developed SAEM data set with three existing data sets: RSEG, the data set from Riggs et al. (2023), and RSQ. (a) Spatial distribution of the gauges in RSEG in grey and the gauges common with SAEM in green. (b) Same as (a) but between SAEM and Riggs et al. (2023).(c) Same as (a) but between SAEM and RSQ. (d) CDF of the KGE values for the common gauges between SAEM and RSEG with simultaneous in-situ data. (e) Same as (c) but between SAEM and Riggs et al. (2023). (f) Same as (c) but between SAEM and RSQ.

discharge estimates with high accuracy (high KGE and very low RMSE) can serve as prior estimates for the SWOT satellite
mission, which aims to provide global discharge estimates. In SAEM, we provide a catalog for each gauge that includes
information such as the distance to the gauge, availability of water level data in the Level-3 databases, and a flag indicating its
contribution to the final discharge estimates. Such information can be used further for optimizing the calibration and validation
processes of hydrological models, enhancing the quality of predictive analytics, and guiding targeted maintenance or upgrades
to existing monitoring infrastructure.

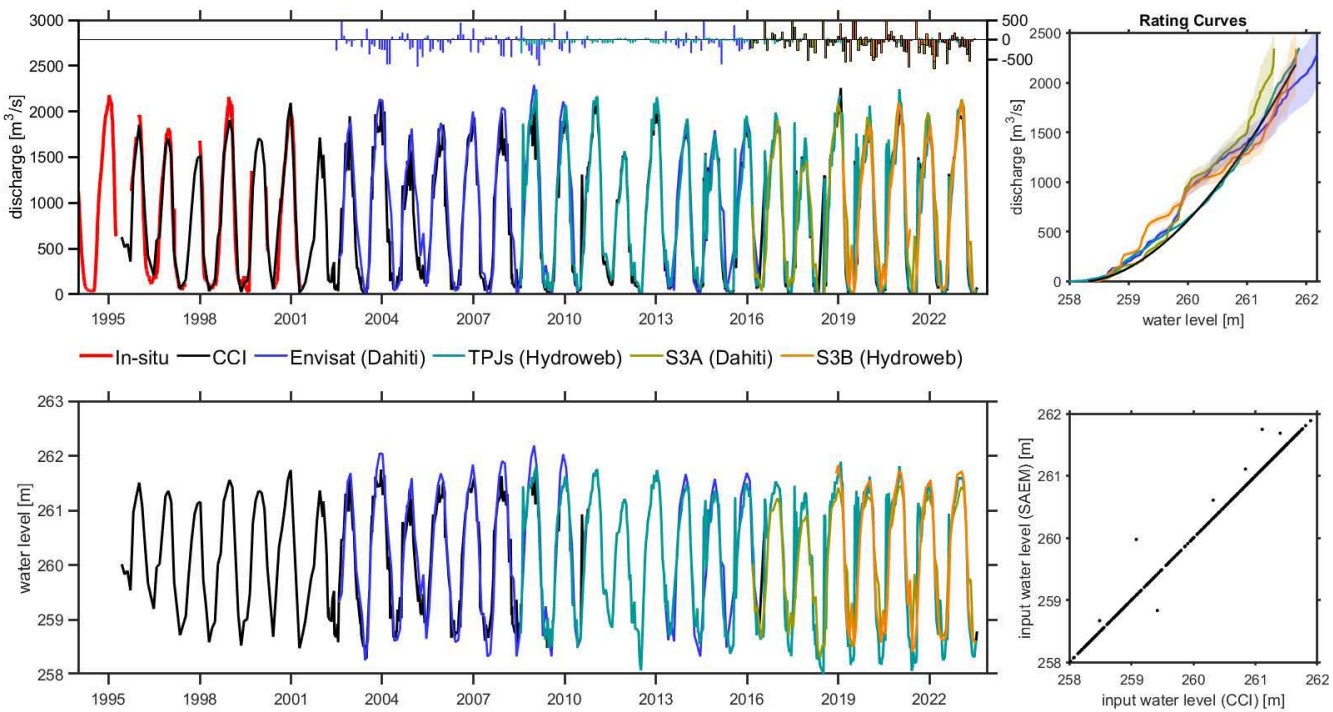

**Figure 10.** Comparison of discharge and water level time series between SAEM and the CCI project for the Niger River. (Top-left) Discharge time series from SAEM's four mission series and CCI, alongside in-situ measurements, with the (CCI - SAEM) difference shown for each orbit family in the top with bar plot. (Top-right) Rating curves were derived from each time series using non-parametric (SAEM) and parametric (CCI) approaches. (Bottom-left) Water level time series from SAEM and CCI, with the shaded region representing the uncertainty of discharge for the SAEM rating curves. (Bottom-right) Scatter plot showing the correlation between water levels in SAEM and CCI data sets. The consistency across data sets highlights the reliability of satellite-derived measurements for river discharge estimation.

For many applications, such as drought characterization, a continuous data set over a long period (more than 30 years) is needed. SAEM includes extended discharge estimates for 354 gauges, which feature at least 30 years of data and gaps of less than 3 years from 1991 to 2020 and have estimated discharge for all months of the year 2021 over the continental United States (CONUS). Here, we use these gauges to explore climate change through continuous gauge records. Figure 11 shows the deviation of the mean annual discharge in 2021 from the mean annual discharge calculated for 1991-2020 at the selected

gauges. To better understand the results, we categorized the deviations into five groups: discharge conditions for 2021 are classified as *much below normal* if they fall below the $10^{th}$ percentile, *below normal* if they are between the $10^{th}$ and $25^{th}$ percentiles, 'normal' if they fall between the $25^{th}$ and $75^{th}$ percentiles, *above normal* if they are between the $75^{th}$ and $90^{th}$ percentiles, and *much above normal* if they are at or above the $90^{th}$ percentile. The percentiles are calculated from the long-term annual discharge time series (1991–2020). The analysis of 354 gauges across the CONUS for 2021 reveals diverse hydrological

responses to climate variability. Over half of the gauges (about 55%) showed *normal* discharge levels, indicating that many

regions maintained hydrological stability in 2021 compared to the past three decades. However, approximately 15% of gauges fell into the *much below normal* category, and another 18% were *below normal*, pointing to significant areas with lower-than-average water availability—likely linked to drought or reduced water supply. Conversely, about 8% of gauges recorded *above normal* discharge, and 5% were *much above normal*, suggesting increased rainfall or other factors driving higher runoff in those areas.

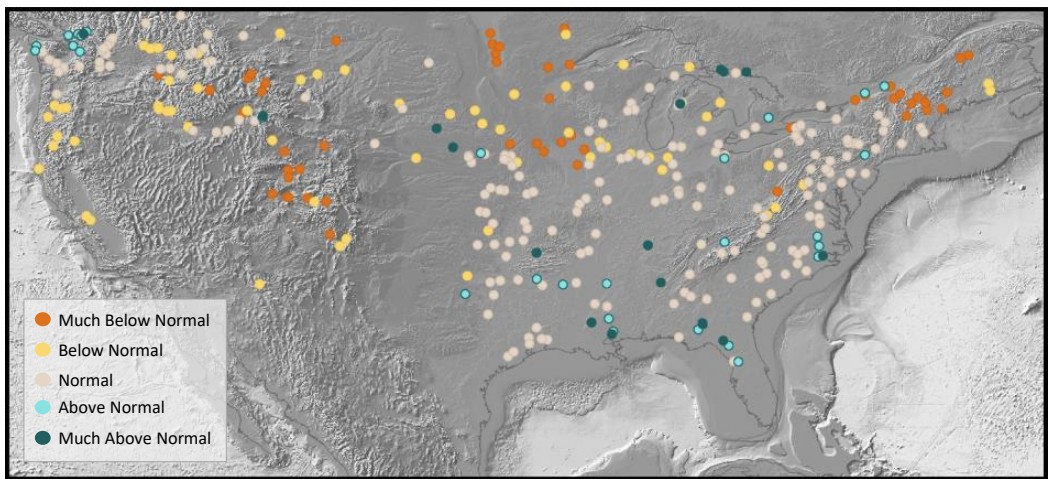

**Figure 11.** Geographic distribution of 2021 discharge deviations among 354 SAEM gauges across the continental U.S., each with at least 30 years of data and minimal gaps from 1991 to 2020, showing variations from much below normal to much above normal relative to the historical average.

## 5.3   Discussion

While the SAEM data set offers a comprehensive suite of products, including estimated discharge, a catalog of altimetry VS, rating curves, and water levels at VSs, it is important to address the inherent limitations of satellite altimetry, particularly when applied to riverine environments. Altimetry can be challenging over narrow rivers due to the difficulty in accurately detecting and tracking the water surface within the confines of the riverbanks (Calmant and Seyler, 2006). Such a challenge is exacerbated during the summer months when reduced water levels and increased vegetation can further complicate signal retrieval. Additionally, the challenges posed by ice-covered river surfaces and the difficulty in deriving reliable time series limit the availability of altimetric water level data at high latitudes, resulting in sparse coverage in these regions (Berry et al., 2005). Despite these challenges, the inclusion of a catalog of virtual stations in the SAEM data set holds significant value. This catalog provides a foundation for developing dedicated retracking algorithms tailored to specific riverine conditions, potentially mitigating some of the limitations. By refining these algorithms, it is possible to enhance the accuracy of water

level measurements and discharge estimates, thereby improving the utility of satellite altimetry for hydrological studies and water resource management (Papa et al., 2010b).

Another significant limitation of satellite altimetry is its temporal sampling. Satellite altimetry missions often provide coarse-sample measurements in time, which can be insufficient for capturing the dynamics of river discharge, especially during rapid hydrological events. Densification algorithms, as demonstrated by (Tourian et al., 2016; Boergens et al., 2019; Nielsen et al., 2022), can partially mitigate this limitation by improving the temporal resolution of discharge estimates. However, capturing rapid events on the order of a few days remains challenging, as this would require sampling frequencies significantly higher than those achievable by current satellite altimetry missions (Cerbelaud et al., 2024). Despite this, integrating densification methods with the SAEM data set still provides a valuable pathway for generating dense discharge time series and addressing temporal sampling limitations for broader hydrological applications.

Further, we acknowledge the assumption of stationarity made in our discharge estimation method, which could lead to extra uncertainty in discharge especially when simultaneous data of water level and discharge are not available. Such an assumption implies that the statistical properties of the relationship between water level and discharge do not change over time, which may not always hold in dynamic river systems. Variations in climatic conditions, land use, and river morphology can all influence this relationship, potentially introducing errors into our discharge estimates (Tourian et al., 2013). Recognizing this limitation is crucial for interpreting our results and underscores the need for developing more robust methods that can account for non-stationary conditions in hydrological studies. Future research should focus on incorporating adaptive algorithms and additional environmental variables to improve the accuracy and reliability of discharge estimations.

While the non-parametric method for discharge estimation faces challenges related to stationarity, it offers a useful alternative to traditional parametric rating curve models. Parametric models, which are grounded in hydraulic principles, often rely on predefined functional forms. While this ensures physical interpretability, such models may underfit when these functional forms fail to capture the inherent variability and complexity of river channels (Elmi et al., 2021; Kirchner, 2006). In an ideal scenario, modeling such behavior would require a full-dimensional process representation based on a comprehensive understanding of the processes, their heterogeneity, and their spatio-temporal dependencies. However, this is rarely feasible in practice, and missing dimensions or physics often lead to mismodeling that propagates and reduces the accuracy of parametric approaches for obtaining rating curves (Gharari and Razavi, 2018). In contrast, the non-parametric method, used in this study, offers greater flexibility by allowing the data itself to define the relationship between water levels and discharge. This adaptability can result in more accurate and reliable estimations, particularly in capturing localized and complex dynamics (Elmi et al., 2021). However, non-parametric methods are not without limitations. They can be prone to overfitting, especially in the presence of localized anomalies or when working with low-quality input data.

## 6  Conclusions

River discharge serves as a vital metric, capturing the volume of water passing through a river cross-section at any given moment. However, existing river discharge data sets face several challenges, particularly due to the decreasing number of

operational gauges. To address that, we have developed the Satellite Altimetry-based Extension of global-scale in situ river discharge Measurements (SAEM v1.1). We have assessed 47 000 gauges worldwide and obtained discharge estimates for 8 730 gauges, more than the existing data sets. In the following, we summarize the benefits of the SAEM data set:

1. SAEM utilizes the multi-satellite altimetry missions and estimates the discharge using the existing global network of national and international gauges.

2. In addition to providing extended river discharge measurements and their uncertainties, SAEM delivers a catalog of Virtual Stations (VSs) for each gauge. This catalog forms the backbone of the SAEM data set. The catalog, along with the gauge coordinates, provides information about the VSs around each gauge, including their satellite mission, track number, distance to the discharge gauge, and a flag indicating whether the VS is used for estimating discharge.

3. Furthermore, SAEM includes water level time series generated specifically for this study, as well as the ID and informa-
tion of water level time series from Level-3 databases. The Level-3 data are sourced from pre-existing databases (more than 40 000 VSs) including Hydroweb, the Database of Hydrological Time Series of Inland Waters (DAHITI), the Global River Radar Altimeter Time Series (GRRATS), HydroSat, and the data set developed by (Kitambo et al., 2022a).

4. The transformation of water level time series into discharge data is modeled through rating curves, derived using a Nonparametric Stochastic Quantile Mapping Function approach developed by Elmi et al. (2021). SAEM delivers rating
curves (~22700 rating curves) for a selected set of VSs, tailored for each VS and mission separately.

Validation against the in-situ data shows that the majority of the KGE values are positive with more than 40% of the cases exhibiting KGE>0.4, Corr.>0.5, and NRMSE<15%. We assessed also the estimated discharge for each orbit family by comparing it with in-situ observations. Based on KGE values, the best performance belongs to Sentinel 3A (median = 0.44), followed by the Envisat series (median = 0.35), Sentinel 3B (median = 0.27), and Topex/Jason series (median = 0.26).
Furthermore, the SAEM discharge time series are compared with three other global-scale discharge data sets, RSEG, Riggs et al. (2023), and RSQ, along with the CCI project. SAEM generally performs similarly to or better than RSEG across 2085 common gauges and shows higher accuracy than Riggs et al. (2023) (in 1926 out of 2168 shared gauges) and than RSG (in ~800 out of 1580 shared gauges)). The comparison with CCI further highlights the reliability of SAEM's non-parametric approach, which effectively captures the water level-discharge relationship. Looking ahead, with the advent of the SWOT
mission as a new tool for discharge estimates globally, SAEM has the potential to serve as a benchmark product for globally assessing discharge estimates, particularly as it continues to be extended and refined.

## 7  Data availability

The SAEM data set is openly available on DaRUS, the data repository of the University of Stuttgart (Saemian et al., 2024) (https://doi.org/10.18419/darus-4475). The following gauge databases are publicly accessible: ArcticNet (www.r-arcticnet.sr.
unh.edu/v4.0/AllData/index.html), Australian Bureau of Meteorology (www.bom.gov.au/waterdata/), Chile Center for Climate

and Resilience Research (https://explorador.cr2.cl/), Canada National Water Data Archive (www.canada.ca/en/environment\
protect\penalty\z@-climate-change/services/water-overview/quantity/monitoring/survey/data-products-services/national-arch\
protect\penalty\z@ive-hydat.html), Brazil National Water Agency (www.snirh.gov.br/hidroweb/serieshistoricas), the Global
Runoff Data Centre (https://portal.grdc.bafg.de/applications/public.html?publicuser=PublicUser), India Water Resources In-
formation System (https://indiawris.gov.in/wris/#/RiverMonitoring), Spain Annuario de Aforos (https://datos.gob.es/en/), Thai-
land Royal Irrigation Department (http://hydro.iis.u-tokyo.ac.jp/GAMET/GAIN-T/routine/rid-river/disc_d.html), Japanese Wa-
ter Information System (www1.river.go.jp/), and the U.S. Geological Survey (https://waterdata.usgs.gov/nwis/rt). The Chinese
Hydrology Project data is not publicly available and was provided by the authors of the data set (Henck et al., 2010; Schmidt
et al., 2011). The GRADES hydrological model outputs are publicly available online (www.reachhydro.org/home/records/
grades).

In our research, we utilized Level-3 databases of water levels sourced from several key repositories. The databases were
acquired from Hydroweb operated by CNES, which provided 24,042 virtual stations (VSs) (see http://hydroweb.theia-land.fr);
DAHITI managed by Deutsches Geodätisches Forschungsinstitut (DGFI), contributing 9,968 VSs (available at https://dahiti.
dgfi.tum.de); HydroSat from the Institute of Geodesy at the University of Stuttgart, offering 2,036 VSs (accessible via http:
//hydrosat.gis.uni-stuttgart.de); and GRRATS supported by Copernicus, European Commission, ESA, USGS, and Amazon
Web Services, providing 1,869 VSs (details at https://blue-dot-observatory.com). Additionally, the Congo basin database,
described in Kitambo et al. (2022), contributed 1,272 VSs (found at https://hess.copernicus.org/articles/26/1857/2022/).

## 8 Competing interests

The contact author has declared that none of the authors has any competing interests.

*Acknowledgements.* Omid Elmi is supported by the DFG (Deutsche Forschungsgemeinschaft) (project Number: 324641997) within the
framework of the Research Unit 2630, GlobalCDA: understanding the global freshwater system by combining geodetic and remote sensing
information with modeling using a calibration/data assimilation approach (https://globalcda.de). Fabrice Papa and Benjamin Kitambo are
supported by the project CNES-TOSCA project DYBANGO. The authors would like to thank Mr. Masoud Jalali Jirandehi, Mr. Pouria
Khadem Haghighat, Mr. Shaurya Prakash, Ms. Jordan Lavey, and Ms. Torian Cooper for their careful visual inspection of the results to
ensure the quality of the data.

## Appendix A

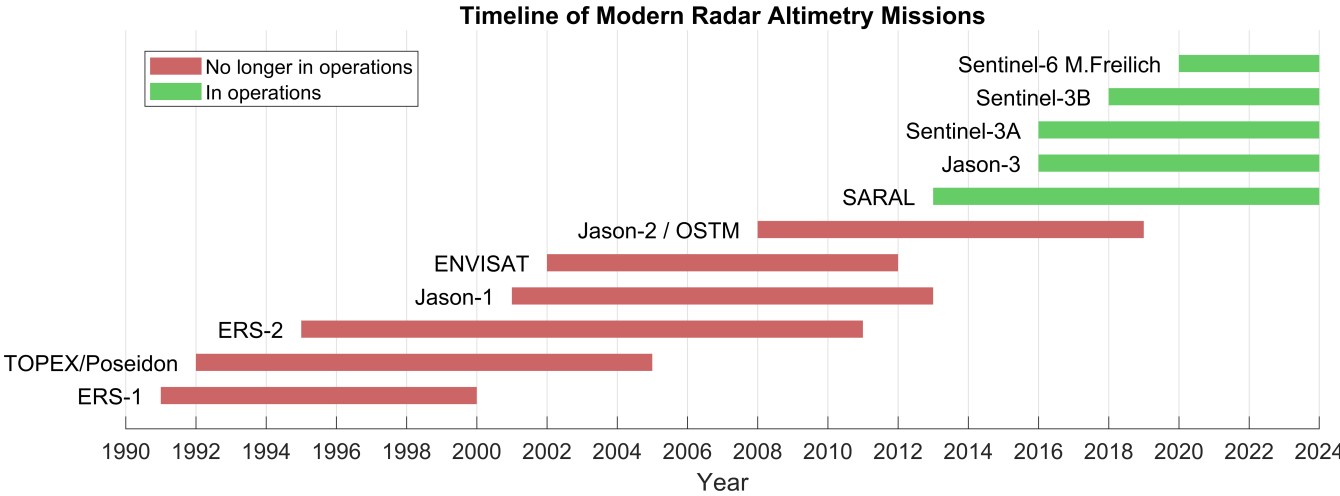

**Figure A1.** The timeline depicts satellite altimetry missions, highlighting operational (in green) and non-operational (in red) periods.

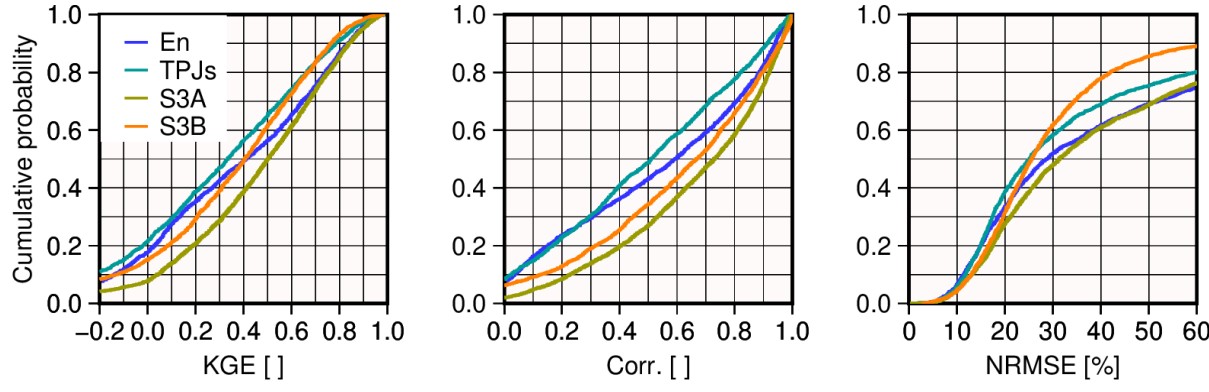

**Figure A2.** Comparison of SAEM discharge estimates with in-situ measurements across four orbit families—Envisat series (En), TOPEX/-Jason series (TPJs), Sentinel 3A (S3A), and Sentinel 3B (S3B)—evaluated using Kling-Gupta Efficiency (KGE), Correlation Coefficient (Corr.), and Normalized Root Mean Square Error (NRMSE).

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
