# Peer review of "Satellite Altimetry-based Extension of global-scale in situ river discharge Measurements (SAEM)"

_Earth System Science Data, 2024_

## Author Comment (AC2)

**RC1: 'Comment on essd-2024-406', Dr. Adrien Paris**

Please find here after my comment on essd-2024-406 manuscript. Overall, the manuscript is well written and easy to read. The dataset presented here will be useful for a large panel of users and my recommendation would be that the manuscript is accepted after revisions.

Some questions remain regarding the possible discrepancies within the dataset not being acknowledged (intermission biases, local differences due to geoid, retracking methods) in this version, regarding the way uncertainties are taken into consideration and regarding some methodological choices. The results also should be further criticized on the light of the input data, given that they may depend on the validation method and/or validation data availability.

Hereafter are my main remarks with the corresponding lines in the initial version.

Kind regards.

Dear Dr. Paris,

Thank you for taking the time to review our manuscript and for your constructive feedback. We greatly appreciate your recognition of the manuscript's clarity and the potential usefulness of the dataset for a wide range of users.

We have carefully addressed each of your points (inter-mission biases, uncertainties, methodological choices, and the need for a more critical discussion of the results in light of the input data) in the revised manuscript and hope that the changes align with your suggestions.

In the following, you can find our detailed responses to your comments.

Thank you once again for your valuable insights and kind recommendation.

Kind regards,

Peyman Saemian on behalf of all the co-authors

Section 2.3.1

I miss a § at the end of 2.3.1 stating the differences of the aforementioned databases (open accessibility or not, timeliness, NRT availability or not, etc.), so that the reader understands why several databases were used; This could be done extending Table 2.

Has any inter-validation been performed when/where overlaps (in VSs) were found? Are there any in the literature? In this section, the reader should understand what were the choices made in case of overlap and why such choices were made. This is crucial for people that would like to duplicate / extend the dataset.

Thank you for your comment. The main reason for using various databases was to benefit from all publicly available Level-3 products and reduce the load for generating new time series which could be a huge task given the considerable number of virtual stations (VSs) in this study. Regarding accessibility, all databases used are publicly available or upon request. In terms of near-real-time (NRT) data, Hydroweb.Next offers NRT data. For overlaps in VSs between databases, we retained all and included the information in the VS catalog for transparency. For the final discharge estimates, we selected the product that passed our quality control and had the better KGE value.

We added the following text to the revised manuscript and the end of section 2.3.1:

*"The primary aim of using these databases is to benefit from the already available Level-3 products and reduce the computational load of processing water level (WL) time series for virtual stations (VSs). In SAEM, we directly utilize the quality-controlled WL time series provided by these databases without any reprocessing or post-processing. Regarding accessibility, all databases are publicly available or upon request. In terms of near-real-time (NRT) data, Hydroweb.Next offers NRT data. When overlaps in VSs were identified between databases, we retained all products in the VS catalog for transparency. The product that passed quality control and achieved better statistical metrics was selected for the final discharge estimates."*

Section 2.3.1 §1 Hydroweb.next

- Please check the satellites list (L97-98) which were used to produce Hwb-Next time series of rivers WL;

  Thank you for bringing this to our attention. We have reviewed and revised the list of satellites in the revised manuscript.

- The data sources (L99,100) and the reference apply for lakes and not rivers; For rivers, please cite Santos da Silva J., S. Calmant, O. Rotuono Filho, F. Seyler, G. Cochonneau, E. Roux, J. W. Mansour, Water Levels in the Amazon basin derived from the ERS-2 and ENVISAT Radar Altimetry Missions, Remote Sensing of the Environment, 2010, doi:10.1016/j.rse.2010.04.020

  Thank you for pointing this out. We have revised the manuscript to replace the reference with the appropriate citation for rivers, i.e., Santos da Silva et al. (2010).

Section 3.1 Construction of VS catalog

- L138-139: please specify the dams and reservoirs database used in input

  We have used the reach-wise flag of dams and reservoirs from the SWORD dataset to identify the presence of dams and reservoirs along river reaches.
  The following sentence is added to the revised version:

  *"The presence of dams and reservoirs was determined based on the reach-wise flags provided in the SWORD v16 dataset."*

- Specify that SWORD V16 (from Fig. 3) was used. Is it version dependent, or did you created a customized one, possibly with connectivity issues corrected? Will the code for creating this VS catalog with connectivity and hydrological constraints be made available to the community together with the database?

  Thank you for your comment. We have added the SWORD version (v16) to the caption of Figure 3 and mentioned it in the manuscript (e.g., sub-section 2.1 and sub-section 3.1). The results of SAEM are dependent on SWORD v16 for this version. Connectivity issues in SWORD are minimized by excluding gauges on narrow rivers (Q<10 m3/s) in this dataset. Future versions or improved SWORD datasets could further reduce missed VSs or gauges. Regarding the code, we do not plan to make it publicly available at this stage. We appreciate your understanding.

- L142-143: put the ENVISAT series list in chronological order

  Thank you for pointing this out. We have corrected the mission list to reflect the chronological order: ERS1, ERS2, Envisat, Envisat Extended, and Saral/AltiKa.

Section 3.2 WL time series

- Since SAEM includes L3 WL time series, I consider a dedicated paragraph on inter-mission biases is mandatory. Even though discharges are estimated through mono-mission non parametric curves, such biases shall be taken in consideration by whoever want to use the L3 WL TS from this database or any other all together to build long TS. Moreover since different retrackers are used among the missions.

  Thank you for raising this important point. As you correctly mentioned, inter-mission biases are not directly relevant to the discharge estimates provided in SAEM. At the same time, we agree that discussing these biases is essential to ensure that users are aware of their implications. Additionally, we emphasize that users should be cautious when using the rating curves provided, as they are specific to the retrackers and setups used in the L3 databases or SAEM WL. We have added the following paragraph in the revised manuscript section 4.2 to address these issues:

  *"Although SAEM provides mono-mission discharge estimates using non-parametric rating curves, it is important to acknowledge inter-mission biases that may arise when comparing or combining water*

*level time series (WL TS) from different satellite missions. Such biases, resulting from differences in satellite orbits, calibration, and instrument characteristics, can impact the continuity and consistency of long-term WL TS. Additionally, the use of different retrackers can also introduce biases. In SAEM, the water level time series are specific to the retrackers and processing setups used for each mission. Users who aim to build long-term WL TS by combining Level-3 data from multiple missions should account for these biases to ensure meaningful comparisons."*

- L152-153: this masking choice implies, for past missions, to keep a very low number of "Hᵢ" measurements from your raw altimetric data. For small rivers, this would mean keeping only the one measurement geolocated over the water. What are your statistics on this point (percentage of dates with <2 Hᵢ measurements for example), and do this have an influence on the final median that you process (and which uncertainty do you provide in this case)? I believe this should be further explained.

Thank you for your comment. We have reviewed our data processing to address your concerns. Our analysis indicates that epochs with fewer than two valid "Hi" measurements occur in only 0.6% of the total data epochs, which is consistent across orbit families: approximately 0.2% in the Envisat series, 0.8% in the Topex/Jason series, 0.2% in Sentinel-3A, and only one case in Sentinel-3B. Figure R1 demonstrates the number of dates with only one valid measurement relative to the mean discharge of the gauges, shown in total and for each orbit family separately. As expected, these cases are more frequent in narrower rivers but remain insignificant in terms of the total number of epochs.

For epochs with only one valid "Hi" measurement, we retain the measurement as it represents an aggregated value derived from multiple echoes. With the 75% water occurrence threshold applied, the level of noise is significantly reduced, and we consider the resulting water level estimate reliable. We have added the following explanation to the revised manuscript for clarity:

*"The use of the GSW mask helps maintain the quality of the extracted time series by excluding non-water reflections. Cases with fewer than two valid measurements per epoch are rare (0.6\% of all data epochs) and are retained."*

[Figure]

Fig R. 1. Number of dates with only one measurement relative to the mean discharge of the gauges, shown in all VSs and for each orbit family separately.

- L171: Is XGM2019 used in one of the L3 databases used in this study? Differences between gravity field models can lead to lat/lon dependent biases between the series from external databases (e.g. Hydroweb.next) that are on EGM2008 and the L3 processed time series. In a matter of uniformity, using the same GGM would be recommendable.

We selected the XGM2019e model because it performs better in regions with limited in-situ gravity data, as demonstrated by Zingerle et al. (2020).  Since Dahiti and Hydroweb.Next already use different geoid models, achieving uniformity across databases is not feasible. However, since all these global models are freely available, users can easily retrieve geoid heights for their preferred model using lat/lon coordinates and adjust the orthometric height accordingly.  We have included the following explanation in the revised manuscript to clarify the reasoning behind the choice of the geoid model:

"The geoid height *N* is determined using static gravity field models, specifically referencing XGM2019e (Pail et al., 2018), *which has been shown to perform better in regions with limited in-situ gravity data (Zingerle et al. 2020). Achieving uniformity across databases is not feasible, as different Level-3 water level databases, such as Dahiti and Hydroweb.Next, already use different geoid models. However, since all these global models are freely available, users can retrieve geoid heights for their preferred model using lat/lon coordinates and adjust the orthometric height accordingly."*

Zingerle, P., Pail, R., Gruber, T., & Oikonomidou, X. (2020). The combined global gravity field model XGM2019e. *Journal of geodesy*, *94*(7), 66.

- L180-184: it would be worth having a table with those statistics. I.e. how many TS were generated, how many (in %?) passed the QC check (total and by mission), and why.

Thank you for your comment. We agree that including statistics in the text will enhance clarity and provide a better understanding of the quality control process. To be transparent, it is challenging to trace back the specific reasons behind each rejection, as our code evaluates multiple criteria and automatically flags time series for rejection or acceptance. We have revised the manuscript to incorporate more details directly into the text (sub-section 3.2):

*"In total, 3763 WL time series were generated for 1702 gauges across various orbit families, including 1598 from the Envisat orbit family, 990 from the Topex/Jason orbit family, 561 from Sentinel-3A, and 614 from Sentinel-3B. During the quality control process, 632 WL time series were rejected, representing approximately 17% of the total. Rejection rates varied across orbit families, with 494 WL time series rejected from the Envisat orbit family, 34 from Topex/Jason, 64 from Sentinel-3A, and 40 from Sentinel-3B."*

3.3 Non parametric rating curve

- L187-193: I suggest discriminating the advantages as a function of the approach. Indeed,

Thank you for your comment. We have updated the list to clarify how our algorithm addresses the mentioned limitations. Below is the modified version of the statements in lines 191-193.

- the "does not necessitate simultaneous ..." is not due to the non-parametric and could be also the case for linear regression;
*"it does not require simultaneous gauge-based and space-based measurements, since the algorithm determines the water height-discharge model by matching the quantile functions,"*

- "it assumes no specific predefined ..." is only for the non-parametric, and it should be stated why this is an advantage comparing to other formulations;
*"it follows a data-driven, nonparametric approach rather than relying on predefined linear or power-law relationships to minimize the possibility of mismodeling,"*

- realistic uncertainty: is not dependent on the formulation but comes from the MC simulations and the creation of a stack of WL and Q.
The methodology paper by Elmi et al. (2021) extensively discusses the discharge uncertainty estimates using this method. In this context, "realistic uncertainty" refers to the discharge uncertainty estimates for each discharge percentile caused only by the uncertainty of the input data. So, it is somehow related to the nonparametric nature of the algorithm. The text in the paper will be modified as follows:

*"it provides input-driven discharge uncertainty estimates for each discharge percentile separately rather than relying only on a variance-covariance matrix for model parameters."*

Elmi, O., Tourian, M. J., Bárdossy, A., & Sneeuw, N. (2021). Spaceborne river discharge from a nonparametric stochastic quantile mapping function. *Water Resources Research*, *57*(12), e2021WR030277.

- L206-208: This means that the uncertainty raisen from previous steps is not being used. Why this choice? The "10% multiplicative uncertainty" should be better explained. Also, a sensitivity analysis of the MC algorithm to the uncertainty (and consequently, the sensitivity of the non-parametric RCs derived) shall be investigated (testing different bounds for uncertainty, and even informative uncertainty such as the one coming from WL processing).

  The 10% multiplicative uncertainty is only applied in the first iteration as an initial value because no reliable uncertainty estimates exist for gauge discharge. We chose not to use the uncertainties provided with the altimetric water levels, as these uncertainties differ in definition across data centers, making their integration inconsistent. After the first iteration, the algorithm re-calibrates the input data uncertainties based on the model's performance evaluation. This iteration process continues until the algorithm meets the convergence criteria. We have updated the manuscript to better explain this choice and clarify the 10% multiplicative uncertainty:

  "In the initial iteration, the algorithm considers a multiplicative uncertainty of 10% of the signal for the input time series. *This decision is due to the lack of available uncertainty estimates for the gauge discharge dataset and the inconsistent definitions of uncertainties across altimetric water level databases.* As the algorithm progresses, it refines its estimates by updating the measurement uncertainties at each iteration. This iterative process continues until the termination condition is met; *however, a maximum number of iterations is also set to ensure the algorithm converges within a predefined limit.*"

- L208: Convergence of MC algorithms is an important point. What convergence criteria was considered, and did all the Q/H Ts reached convergence? If not, how many did not passed, and on which reaches, rivers, etc?

  We use a maximum iteration criterion to ensure the Monte Carlo (MC) algorithms converges within a predefined limit (add to the revised text for clarity). Additionally, our quality control steps are designed to mitigate problematic cases where convergence may not be achieved. While this is an important topic, a detailed investigation of non-converging cases is beyond the scope of this study and can be explored further in future work.

5 Results

- L275-279: A statistics on the step where data was rejected could be very important: was it due to a lack of raw (gdr) data, during the WL processing, during the NPQM, quality test? etc. The visualization could be done maybe with a geographical (lat/lon) view?

  Thank you for your comment. We have added detailed statistics in the revised manuscript to provide insight into the rejection process at each step. After applying the mean discharge threshold of greater than 10 m³/s, the number of gauges considered decreased from approximately 47,000 to 15,040.

Among these, 8,730 gauges are included in SAEM v1.0. For the 6,310 gauges not included in SAEM, only 1,405 were within 12 km of a SWORD reach, with the remaining gauges being rejected due to greater distances. Of the 1,405 gauges near SWORD reaches, 671 had water level (WL) data either from Level-3 databases or SAEM WL. For these gauges, the Non-Parametric Quantile Mapping (NPQM) method failed to converge for any orbit family in 69 cases, while it successfully converged for at least one orbit family in 602 cases. Of the 602 cases ultimately rejected, 263 were excluded after visual inspection, and 339 were rejected due to failing statistical thresholds. We added this information to the revised manuscript.

- L300: remove the "exceptionally". The model performs indeed well in these regions, yet KGE and Corr remain in the "good" domain and not in the "exceptional"
  Done!
- L305: "the overall good accuracy" instead of "high accuracy", for the same reasons than above
  Done
- L306: is 73% the right value? Please check, this seems pretty high to me.

  Thank you for catching this error. You are correct, and the correct value is 18%. We have revised the manuscript to reflect this correction at Line 306.

- L310-311: I am quite surprised by the low values show by S3B (same as T/P Jason). It is important to bring some elements of explanation in the discussion section, since S3B is expected to perform at least as good as S3A, at least in terms of WL estimates. What can explain such difference? The length of the TS? The lack of in situ data for validation on the recent period? Anythink else? Please discuss it. Regarding Topex/Jason also, are all the missions giving similar results? Or is the global quality impacted by the oldest missions? When ENVISAT is mentioned, is it only ENVISAT or is it the family (ERS/ENV/SRL)?

  Thank you for your comment. As you correctly mentioned, several factors contribute to the observed differences in performance between Sentinel-3A (S3A) and Sentinel-3B (S3B). One primary reason is the length of the water level time series, with S3A operational since 2016 and S3B since 2018, resulting in a shorter data record for S3B. Additionally, there are more cases with simultaneous water level and discharge data for S3A (4483 out of a total of 8928 cases) compared to S3B (2369 out of 4791 cases). The cumulative distribution function (CDF) plots in Figure A2 shows performance based only on simultaneous cases, which are nearly double for S3A, influencing the comparison. Furthermore, the different orbits of S3A and S3B could also play a role in their performance.
  Regarding the Topex/Jason orbit family, the global quality is likely impacted by the older missions, as these missions span multiple generations with differing sensor capabilities and retracking algorithms. Finally, when referring to Envisat in the manuscript, we mean the Envisat orbit family, which includes data from ERS, Envisat, and Saral/AltiKa.
  We have revised the manuscript to include these considerations and provide further discussion:

  *"The difference in performance between Sentinel-3A and Sentinel-3B is influenced by several factors. Besides the difference in their orbit, Sentinel-3A has been operational since 2016, providing a longer*

*data record compared to Sentinel-3B, which began in 2018. Additionally, there are significantly more cases with simultaneous water level and discharge data for Sentinel-3A compared to Sentinel-3B (4483 vs 2369). For the Topex/Jason orbit family, the global quality is influenced by the older missions, which may exhibit higher uncertainty due to differing sensor capabilities and retracking algorithms."*

- L345-348: provide other metrics to complement KGE (KGE improvement of 0.15 is somehow unease to quantify) (e.g. NRMSE in %, other)

  Thank you for the suggestion. We have added the values of the correlation coefficient and NRMSE (in %) to the revised manuscript.

- L361: CCI WL rely mostly on hand-processed time series using Altis software (and in particular it is the case for Niger TSs, please correct.

  Thank you for pointing this out. We corrected the text in the revised manuscript.

- Figure 10: I would rather show the comparison in terms of anomaly (or also show), in order to better evidence differences between SAEM and CCI for all the discharge amplitude (can be anomaly or normalized anomaly). Differences for small discharges do not appear clearly here. Same for WL.

  Following your recommendation, we have revised Figure 10 to include the differences between SAEM discharge and CCI discharge, color-coded by different orbit families. This representation better highlights the differences across the entire discharge amplitude range, including for smaller discharges. Additionally, we have incorporated the SAEM rating curves along with their corresponding uncertainties. The revised figure shows that the parametric approach used in CCI generally lies within the uncertainty range of SAEM, except for the middle range of quantiles. This difference is explainable, as the non-parametric approach in SAEM is designed to closely follow the behavior of the data, while the parametric approach in CCI maintains a power-law relationship across the entire range of quantiles. We have updated the manuscript and figure accordingly to reflect these changes.

5.3 Discussion

- I miss a paragraph dedicated to the discussion around uncertainties. For example, does the CCI RC fall inside the uncertainty bound for SAEM RCs? What is the contribution of providing uncertainty in discharge estimate. Does this uncertainty is useful, and in line with other (e.g. CCI) provided uncertainties?

  Thank you for your comment. The NPQM method used in our study has been thoroughly presented and discussed in our previous works:
  https://www.nature.com/articles/s41597-024-03078-6
  https://agupubs.onlinelibrary.wiley.com/doi/full/10.1029/2021WR030277

  so we did not revisit it in detail in this manuscript. However, based on your suggestion, we have added a brief mention of uncertainties at the end of the methods section to ensure clarity for readers.

Additionally, we have included the uncertainty values for the rating curves in Figure 10 to enable a comparison of the rating curves (RC) from CCI and SAEM, which is already mentioned and discussed in the previous comment.

- L420-425: This matter is worth discussing a little more. See Kirchner 2006 "getting the right answers", the choice of non-parametric vs parametric RCs can be discussed here, bringing together the advantages (as already shown) but also the drawbacks of both methods (hydraulic-based vs data-fit based)

Thank you for the suggestion and for referencing Kirchner (2006). We agree that a discussion on the advantages and drawbacks of parametric (hydraulic-based) and non-parametric (data-fit based) methods is valuable. We have included the following text to the revised text to include this discussion:

*"While the non-parametric method for discharge estimation faces challenges related to stationarity, it offers a useful alternative to traditional parametric rating curve models. Parametric models, which are grounded in hydraulic principles, often rely on predefined functional forms. While this ensures physical interpretability, such models may underfit when these functional forms fail to capture the inherent variability and complexity of river channels (Elmi et al., 2021, kirchner 2006). In an ideal scenario, modeling such behavior would require a full-dimensional process representation based on a comprehensive understanding of the processes, their heterogeneity, and their spatio-temporal dependencies. However, this is rarely feasible in practice, and missing dimensions or physics often lead to mismodeling that propagates and reduces the accuracy of parametric approaches for obtaining rating curves (Gharari and Razavi, 2018). In contrast, the non-parametric method used in this study offers greater flexibility by allowing the data itself to define the relationship between water levels and discharge. This adaptability can result in more accurate and reliable estimations, particularly in capturing localized and complex dynamics (Elmi et al., 2021). However, non-parametric methods are not without limitations. They can be prone to overfitting, especially in the presence of localized anomalies or when working with low-quality input data."*

Gharari, S., & Razavi, S. (2018). A review and synthesis of hysteresis in hydrology and hydrological modeling: Memory, path-dependency, or missing physics?. *Journal of hydrology*, *566*, 500-519.

- As said above, the relative performance of SAEM discharges as a function of the mission should be discussed, since the classical "quality evolution" with time is not respected here. This can be due to several aspects, which need to be discussed so that the reader/user understand what lies in this validation statistics.

As it is already mentioned in the previous comments, we have revised the text to include a detailed comparison of the performance of SAEM discharge estimates across different orbit families. The revised manuscript also discusses the possible reasons behind the observed performance variations, providing readers and users with a clearer understanding of the validation statistics.

---

## Author Comment (AC3)

**RC2: 'Comment on essd-2024-406', Dr. Arnaud Cerbelaud**

**General comments - Overall quality of the preprint**

This manuscript presents a novel dataset, called Satellite Altimetry-based Extension of the global-scale in situ river discharge Measurements (SAEM), that leverages multiple satellite altimetry missions to extend discharge estimates based on H-Q non parametric rating curves calibrated at in situ gauges. Median KGE is 0.48 for 8730 gauges extended. A catalog is also introduced, providing information (location, distance to gauge, discharge series, water level series, rating curves for the mapping…) on the virtual stations used to obtain good quality discharge close to the gauges.

I believe this manuscript to be of very high relevance to the hydrology and remote sensing communities as it is the very first to attempt creating a global dataset of discharge estimates from altimetry virtual stations alone (where most studies until now have been at continental scale at most, or using imagery-based discharge estimates predominantly). Whatever the accuracy achieved, this is no small feat, and it will be particularly useful in the context of the recent SWOT mission, which should deliver an official discharge product globally. Congratulations to the authors.

It is well written and organized, and I recommend acceptance after (quite a few but) minor revisions.

Best regards,

Dear Dr. Cerbelaud,

Thank you for your thorough review of our manuscript and for your encouraging and supportive comments. We are pleased to hear that you find the dataset and its methodology relevant and valuable to the hydrology and remote sensing communities. Your recognition of the novelty and potential impact of our work, especially in the context of the SWOT mission, is greatly appreciated.

We have carefully considered your detailed comments and suggestions and have addressed them in the revised manuscript. In the following, you will find our responses to each of your points.

Thank you once again for your thoughtful review and recommendation.

Best regards,

Peyman Saemian on behalf of all the co-authors

**Specific comments - Individual scientific questions/issues**

Intro

- l.27: "volume of water …": rather "volumetric flow rate"? you are missing the per unit time information. (same remark in abstract)

  Thank you for your comment. You are correct that "volumetric flow rate" is a more precise term, as it explicitly includes the per unit time information. We have revised the text in both the abstract and manuscript to reflect this definition.

- l.29: is confronted with?

  We changed it into "faces" in the revised manuscript to keep simplicity.

- l.35: I would remove "directly". Hydraulic variables can be rather approximated with complex measurements and processing.
  Done!

- l.35: the seminal paper for water levels is I think Birkett, 1998 (10.1029/98WR00124), similarly to Smith, 1995, 1996 for river widths
  Done!

- l.36-37: and even where no gauge exists, using algebraic flow laws or hydraulic models.

  Thank you for the suggestion. We have revised the text to include the possibility of estimating discharge in ungauged locations using algebraic flow laws or hydraulic models.

- l.37-41: for rating curves, you can cite Leopold and Maddock, 1953, who pioneered the notion of hydraulic geometry relationships, whether from depth, width or speed.
  Done!

- l.42-51: I think you can make for an even stronger argument to your manuscript. There are a few studies that went global on discharge from satellites, but mostly from an imagery standpoint (i.e., using width-based rating curves), e.g., Elmi and Tourian, 2023, Elmi et al. (2024), Riggs et al., 2023, but also Lin et al., 2023 (10.1016/j.rse.2023.113489) that you don't mention. To my knowledge, none have gone global on altimetry only!! So you are the first, that's something.

  Thank you for your comment and for highlighting the study by Lin et al. (2023). We agree that acknowledging prior global-scale efforts strengthens our manuscript. We have included Lin et al. (2023) in our comparison (Sec. 5.1) to provide a more comprehensive overview of existing datasets. Additionally, we have emphasized that SAEM is the first dataset to apply satellite altimetry exclusively at a global scale.

Section 2

- Table 1: N gauges for USGS and Chinese database are the same, typo?

  Thank you for pointing this out. You are right, and upon review, we found that the entire row for the Chinese database was redundant. We have removed it from Table 1 in the revised manuscript.

- l.97-98: to my knowledge Hydroweb.next does not contain ERS-1, TOPEX, Jason-1 nor GFO virtual stations. Please check? Also, CLS company is in charge of retracking for the most part I believe?

  Thank you for the comment. You are correct, and we have excluded ERS-1, TOPEX, Jason-1, and GFO from the text in the revised manuscript. Regarding the processing, we have clarified that the time series are specified by LEGOS and computed by CLS on behalf of CNES, Theia, and the Copernicus Global Land service. The text has been updated accordingly.

- I think you can mention that Hydroweb.next (and DAHITI?) are the only databases currently providing "live" level-3 data? The big advantage of your catalog will be for people to use your rating curves in the future as new WL data comes out.

  Thank you for your comment. You are correct that Hydroweb.next is currently the only databases providing Near Real Time (NRT) Level-3 water level data. As you correctly mentioned, one of the major advantages of our catalog lies in enabling users to apply the rating curves to future water level data as they become available. This point has been highlighted in Section 4 of the revised manuscript.

- Overall a few details on the extent to which level-3 databases differ would be helpful (retracker etc.). What if the same VS is present in different databases, which WL series do you retain?

  While it would indeed be interesting to compare the details of the differences between Level-3 databases, such as the retrackers, this is beyond the scope of our study. The primary aim of using multiple databases was to include as many L3 time series as possible and reduce the workload of generating new time series. Regarding overlaps in Virtual Stations (VSs) between databases, we retained both time series and included this information in the VS catalog for transparency. For the final discharge estimates, we selected the product that passed our quality control and demonstrated a better KGE value. We added the following text to the revised manuscript and the end of section 2.3.1:

  *"The primary aim of using these databases is to benefit from the already available Level-3 products and reduce the computational load of processing water level (WL) time series for virtual stations (VSs). These databases provide quality-controlled WL time series, eliminating the need to reprocess the data independently. Regarding accessibility, all databases are publicly available except the Congo dataset which is available upon request. In terms of near-real-time (NRT) data, Hydroweb.Next offers NRT data. When overlaps in VSs were identified between databases, we retained both products in the VS catalog for transparency. The product that passed quality control and achieved better statistical metrics was selected for the final discharge estimates."*

- Figure 2: it is kind of hard to see all colors (especially Hydrosat in orange), but I don't have a way around that…

  Thank you for your comment. We have revised Figure 2 to improve the visibility of the colors, including Hydrosat. We hope the new version is clearer and addresses your concern.

Section 3

- 3.1: SWORD (being Landsat-based from GRWL) has very few low-order reaches (compared to MERIT that's DEM-based for instance). From experience, a handful of your gauges are located on reaches that are not contained in SWORD. So you do not associate VSs to these gauges?

  Thank you for your comment. Our choice to use SWORD was motivated by several factors. First, SWORD's Landsat-based approach provides higher-resolution representation (30 m) of river widths, capturing river complexity better than DEM-based databases such as MERIT Hydro, which are at coarser resolution (~90 m) and do not always reflect the detailed structure of river networks (Altenau et al. 2021). Additionally, SWORD enables direct connectivity with SWOT mission users, allowing them to benefit from the reach_id information we provide. For the current version of SAEM, we applied a mean discharge condition of Q>10m3/s, which excludes many gauges located on narrower stems or low-order tributaries that might otherwise be missing in SWORD. In future versions of SAEM, we aim to expand coverage to include such gauges and incorporate newer versions of SWORD to address these limitations.

  Altenau, E. H., Pavelsky, T. M., Durand, M. T., Yang, X., Frasson, R. P. D. M., & Bendezu, L. (2021). The Surface Water and Ocean Topography (SWOT) Mission River Database (SWORD): A global river network for satellite data products. *Water Resources Research*, *57*(7), e2021WR030054.

- Figure 3: water occurrAnce typo
  Done!

- Figure 3: Again, what happens if the gauge is located on a small tributary reach not contained in SWORD?

  In cases where a gauge is located on a small tributary reach not contained in SWORD, we exclude that gauge from our assessment.

3.2:
- l.151-153 what about when/if GSW and SWORD don't line up (even though they are both Landsat-based), you don't get any h¡?

  The primary reason for using the GSW occurrence map is to ensure that the altimetry data reflect water surfaces, thereby reducing noise in the extracted time series. Regarding the potential misalignment between GSW and SWORD, we believe this issue is minimal due to the 75% threshold

applied. Cases of misalignment are expected to be rare and negligible in their impact on the results.

- l.160-161: Overall I am not competent enough to review retracking skills, but it would be good to have a little more information on, for instance, what retrackers are used in the level-3 databases compared to what you use for SAEM, and how they differ.

  The Level-3 databases used in this study employ different retrackers depending on their processing strategies, which are often optimized for specific missions or conditions. As mentioned in the response to one of the earlier comments, for SAEM, we utilize the water level time series directly from these databases without reprocessing, as our primary focus is on incorporating as many reliable time series as possible while reducing the workload of generating new data. While a detailed comparison of retrackers between these databases and their potential differences would be valuable, such an analysis is beyond the scope of this study. We added the following text in the end of the sub-section 2.3.1:

  *"In SAEM, we directly utilize the quality-controlled WL time series provided by these databases without any reprocessing or post-processing."*

- l.171: the geoid model chosen differs from, e.g., egm2008, used for other retracking. Does that impact your WL series?

  The choice of geoid model, or the fact being different from the EGM2008 should not be a concern. All these geoid models perform similarly over well-surveyed regions where sufficient gravity data are available. According to Zingerle et al. (2020) the XGM2019e can be considered as a better geoid model, especially in less well-surveyed areas, with limited gravity data. Therefore, we believe this choice ensures better performance and reliability for our application. We have added this information in the revised                                        manuscript                                        for                                        clarity.

  Zingerle, P., Pail, R., Gruber, T., & Oikonomidou, X. (2020). The combined global gravity field model XGM2019e. *Journal of geodesy*, *94*(7), 66.

- l.180-184: the outlier removal procedure is referred to in Tourian et al., 2022, ok. But for the quality control, what does it take to "pass" the control?

  Thank you for your comment. We have added more details to the revised manuscript to clarify the quality control (QC) process. Specifically, we have expanded on how the length of the time series is evaluated with respect to the theoretical number of observations based on the satellite's repeat period. We have also included a brief mention of bias control and other aspects of the QC process:

  *"Finally, quality control is conducted on the generated water level time series to select those with the best quality. This evaluation includes assessing the length of the time series in relation to the theoretical number of observations, which is derived from the satellite's repeat period and the observation duration. The time series are further checked for statistical characteristics, including the distribution of values, skewness, and variability, to ensure they are representative of water level changes. Outliers are*

*identified and limited to a maximum 10% to maintain data reliability. Additionally, bias control is implemented by verifying the consistency of mean and median values across segments of the time series. The alignment of the time series with Digital Elevation Model (DEM) information is also assessed to ensure consistency with expected elevations. In total, 3763 WL time series were generated for 1702 gauges across various orbit families, including 1598 from the Envisat orbit family, 990 from the Topex/Jason orbit family, 561 from Sentinel-3A, and 614 from Sentinel-3B. During the quality control process, 632 WL time series were rejected, representing approximately 17% of the total. Rejection rates varied across orbit families, with 494 WL time series rejected from the Envisat orbit family, 34 from Topex/Jason, 64 from Sentinel-3A, and 40 from Sentinel-3B."*

3.3:

- l.189-190: I would say" This method overcomes several limitations, since:" The fact that it does not necessitate simultaneous measurements of gauges and WL comes from the use of quantile mapping, and is not related to linear regression.
  Thank you. We revised the text to clarify this point.

- l.201-203: what is the 3 sigma test? Could explain a little more. From what I understand in the matlab code, we update at each iteration the perturbation/uncertainty applied to the Monte Carlo simulations to the max of the distance between the estimated/simulated and measured quantiles divided by 3.

  The 3-sigma test is a statistical concept used to determine the range within which most data points lie in a normal distribution. It states that approximately 99.7% of data falls within three standard deviations of the mean. In this algorithm, for re-calibrating the input error uncertainty, we assumed that all the residual values (estimated-measured) should be inside the 3-sigma region. Therefore, we scale the uncertainties for the next iteration to satisfy this condition. The following text is added to include this clarification in the revised manuscript:

  "evaluating the performance of the derived model by comparing the estimated and measured discharge of the evaluation sample performing a 3σ test. If available, the evaluation sample consists of simultaneous gauge- and space measurements. Otherwise, measurements from both data sets within the same quantile are included in the evaluation sample. *The 3σ test is a statistical approach that assumes approximately 99.7% of data falls within three standard deviations of the mean in a normal distribution. In this algorithm, the test ensures that the residuals (estimated - measured discharge) remain within this range,*

  updating the measurement uncertainties with respect to the result of the 3σ test, *scaling them accordingly to maintain consistency in the iterative process,*"

3.4

- l.222-225: not a big fan of these statistical tests as I find they are usually rather easy to pass. I would just trust more KGE and its three components, NBIAS, NSTDERR and Corr.

  Thank you for your comment. While we agree that statistical tests like KS can be relatively easy to pass, we found cases where KGE alone, even with a threshold, did not adequately capture the similarity in distributions. The KS test serves as a complementary measure to ensure that the estimated discharge distribution aligns well with the gauge measurements. This additional step helps us retain datasets with both accurate and representative distributions. We hope this clarifies our approach.

- l.226: case 2 no simultaneous measurements. KGE between what? Monthly mean aggregates? Seems that is the case because that is what you are using for the SW test after.

  Thank you for your observation. You are correct that for Case 2, we computed the KGE between the mean monthly discharge of in-situ and estimated data. We have revised the manuscript to clarify this.

- Visual inspection on so many VSs, what a tedious job!

  Indeed, visual inspection of such a large number of VSs was a labor-intensive task, but it was crucial to ensure the quality and reliability of the dataset. We appreciate your acknowledgment of the effort involved.

- l.232: only 1400 cases rejected out of 43000? I would expect way more?

  Thank you for pointing out this mistake. We have updated the text and the flowchart (Figure 5) with the corrected numbers.

Section 4

- 4.3: so these rating curves are the 100 quantiles for WL and Q mapped from NPQM? So that anyone can interpolate a new WL observation into discharge using the 200 data points?

  Exactly. The rating curves consist of 100 quantiles for WL and Q mapped using NPQM, enabling users to interpolate new WL observations into discharge using the new data points. This approach facilitates generating discharge from new WL observations within the same orbit family and allows the use of SAEM for near-real-time (NRT) discharge estimation when paired with NRT water level time series.

- 4.4: Are the uncertainties given by quantiles in the RCs as well (from the standard dev of the stack of MC simulations?)? So that users can obtain uncertainties for future discharge estimates they compute?

  You are right! Users can estimate uncertainties as well using the quantiles in the rating curves, derived from the standard deviation of the Monte Carlo simulations. Additionally, users can estimate uncertainties for their discharge computations using the uncertainty lookup tables provided alongside

the rating curves, which are derived from the standard deviation of the Monte Carlo simulations. We have added this information to Section 4.3 for clarity.

Section 5

- l.287: "mostly exceeding 0.76": the median (25$^{th}$ perc) corr is 0.65 (0.4) later in Figure 8.

  The phrase "mostly exceeding 0.76" referred to Figure 7, which presents selected cases, whereas Figure 8 shows the comparison across all cases with simultaneous data. To minimize confusion, we have removed the word "mostly" in the revised manuscript.

- l.288 + l.289-290: correlation is important, but I wouldn't say it indicates robust nor reliable performance alone. WL and discharge should be in phase, otherwise one of the two is just wrong. Robust performance to me is also about having very little bias and similar standard error (i.e., all three components of KGE).

  As you correctly pointed out, correlation alone should not be over-interpreted as an indicator of robust or reliable performance, as metrics like KGE and its components provide a more comprehensive evaluation. However, a high correlation does indicate that the estimated and in-situ data follow each other well. Additionally, as mentioned in the response to the previous comment, the correlation coefficients in Figure 7 are presented to illustrate the performance of selected cases, while a more comprehensive comparison with metrics including KGE and NRMSE is provided later in Figure 8. To prevent over-interpretation, we have replaced the term "robust" with "good" in the revised manuscript.

- l.295-296: Is the NRMSE normalized by the mean of obs, the RMS of obs, or by the difference between min and max of obs? Values can vary immensely depending on which is used...

  Thank you for pointing this out. We have normalized RMSE using the standard deviation of the observed discharge, and this information has been added to the revised manuscript for clarity.

- l.306: median NRMSE from the CDF looks more like 18% than 73%!
  You are right. We revised the manuscript.

- l.310-311: Sentinel-3B should perform similarly as Sentinel-3A. So it is probably an issue of WL time series length (S-3A starting 2016, S-3B 2018). Discuss it maybe?

  Thank you for your comment. We agree that the shorter time series of Sentinel-3B compared to Sentinel-3A could influence its performance. We have added the following explanation to the manuscript:

  *"The difference in performance between Sentinel-3A and Sentinel-3B is influenced by several factors. Besides the difference in their orbit, Sentinel-3A has been operational since 2016, providing a longer*

*data record compared to Sentinel-3B, which began in 2018. Additionally, there are significantly more cases with simultaneous water level and discharge data for Sentinel-3A compared to Sentinel-3B (4483 vs 2369)."*

5.3 Discussion:

- l.407-411: Regarding the temporal sampling limitations (10-day revisit at best for Jasons/Sentinel-6), you can say that it is impossible to capture rapid events on the order of a few days (Cerbelaud et al., 2024, found you need to sample 4 times more frequently than the event duration), even with densification algorithms… This means that small, low-order river reaches upstream of basins (which are typically the ones showing rapid changes) cannot be accurately captured. They are too narrow anyways for retracking…

  Thank you for your comment. We agree that capturing rapid events on the order of a few days is challenging, even with densification algorithms, given the current temporal sampling limitations of satellite altimetry missions. We have revised the text in the discussion section to address this point and included a reference to the study you mentioned.

- l.420-425: You could discuss the issue of potential overfitting to data in NPQM versus underfitting in parametric RC (e.g., when a two-step $a(H-H_0)^b$ RC is needed) in light of the comparisons described in 5.1.

  Thank you for pointing this out. We have revised the manuscript and added the following text to address your comment:

  "While the non-parametric method for discharge estimation faces challenges related to stationarity, it offers a useful alternative to traditional parametric rating curve models. Parametric models, which are grounded in hydraulic principles, often rely on predefined functional forms. While this ensures physical interpretability, such models may underfit when these functional forms fail to capture the inherent variability and complexity of river channels (Elmi et al., 2021, Kirchner 2006). In an ideal scenario, modeling such behavior would require a full-dimensional process representation based on a comprehensive understanding of the processes, their heterogeneity, and their spatio-temporal dependencies. However, this is rarely feasible in practice, and missing dimensions or physics in hydrological models often lead to mismodeling errors that propagate and reduce the accuracy of parametric approaches (Gharari and Razavi, 2018). In contrast, the non-parametric method used in this study offers greater flexibility by allowing the data itself to define the relationship between water levels and discharge. This adaptability can result in more accurate and reliable estimations, particularly in capturing localized and complex dynamics (Elmi et al., 2021). However, non-parametric methods are not without limitations. They can be prone to overfitting, especially in the presence of localized anomalies or when working with low-quality input data."

Gharari, S., & Razavi, S. (2018). A review and synthesis of hysteresis in hydrology and hydrological modeling: Memory, path-dependency, or missing physics?. *Journal of hydrology*, *566*, 500-519. Kirchner, J. W. (2006). Getting the right answers for the right reasons: Linking measurements, analyses, and models to advance the science of hydrology. *Water resources research*, *42*(3).

Conclusion

- The conclusion on which mission performs best begs for an explanation. Why does ENVISAT work better than the Jason series when ENVISAT has a 35-day revisit, and Jason a 10-day revisit…? The range window size of ENVISAT? To me Jason-2/Jason-3 should provide so much better results than ENVISAT.

  Thank you for your comment. You are correct that the difference in performance between the Envisat and Jason series missions deserves further explanation. While Jason series missions generally perform better than Envisat, in our analysis, the performance of the Jason orbit family is influenced by its older missions, such as Topex-Poseidon, which may exhibit higher uncertainties due to sensor limitations and earlier retracking algorithms. Based on Figure A2, the Topex/Jason orbit family shows a lower correlation compared to the Envisat orbit family but performs better in terms of NRMSE. This indicates that while the Jason series achieves smaller errors in discharge magnitude, its ability to capture temporal variability is slightly weaker. We have added the following explanation to the revised manuscript (before sub-section 5.1) to address this point:

  *"For the Topex/Jason orbit family, the global quality is influenced by the older missions, which may exhibit higher uncertainty due to differing sensor capabilities and retracking algorithms. Based on Figure A2, the Topex/Jason orbit family shows a lower correlation compared to the Envisat orbit family but performs better in terms of NRMSE. This indicates that while the Jason series achieves smaller errors in discharge magnitude, its ability to capture temporal variability is slightly weaker."*